# DISTRIBUTIONALLY ROBUST RECOURSE ACTION

## ABSTRACT

Recourse actions aim to explain a particular algorithmic decision by showing one or multiple ways in which the instance could be modified to receive an alternate outcome. Existing recourse recommendations often assume that the machine learning models do not change over time. However, this assumption does not always hold in practice because of data distribution shifts, and in this case, the recourse actions may become invalid. To redress this shortcoming, we propose the Distributionally Robust Recourse Action framework, which generates a recourse action that has high probability of being valid under a mixture of model shifts. We show that the robust recourse can be found efficiently using a projected gradient descent algorithm and we discuss several extensions of our framework. Numerical experiments with both synthetic and real-world datasets demonstrate the benefits of our proposed framework.

## 1 INTRODUCTION

Post-hoc explanations of machine learning models are useful for understanding and making reliable predictions in consequential domains such as loan approvals, college admission and healthcare. Recently, recourse is rising as an attractive tool do diagnose why the machine learning models have made a particular decision for a given instance. Recourse work by providing possible actions to modify a given instance to receive an alternate decision (Ustun et al., 2019). Consider, for example, the case of loan approvals in which a credit application is rejected. The counterfactual will offer the reasons for rejection by showing what the application package should have been to get approved. A concrete example of a counterfactual in this case may be "the monthly salary should be higher by $500" or "20% of the current debt should be reduced".

Recourses have a positive, forward-looking meaning: they list out the recourse actions that a person should implement so that they can get a more favorable outcome in the future. If a specific application can provide the negative outcomes with recourse actions, it can improve the user engagement and boost the interpretability at the same time (Ustun et al., 2019; Karimi et al., 2021). Explanations thus play a central role in the future development of human-centric machine learning.

Despite its attractiveness, providing recourse for the negative instances is not a trivial task. For real-world implementation, designing a recourse needs to strike an intricate balance between conflicting criteria. First and foremost, a recourse action should be feasible: if the prescribed action is taken, then the prediction of a machine learning model should be flipped. At the same time, a framework for generating recourse should minimize the cost to take recourse actions to avoid making a drastic change to the characteristics of the input instance. An algorithm for finding recourse must make change to only features that are actionable, and should leave immutable features (relatively) unchanged. For example, we must consider date of birth as an immutable feature; in contrast, we can consider salary or debt amount as actionable features.

Various solutions has been proposed to provide recourses for a model prediction (Karimi et al., 2021; Stepin et al., 2021; Artelt & Hammer, 2019). For instance, Ustun et al. (2019) used an integer programming approach to obtain actionable recourses, and also provide a feasibility guarantee for linear models. Karimi et al. (2020) proposed a model-agnostic approach to generate nearest counterfactual explanations and focus on structured data. Dandl et al. (2020) proposed a method which finds counterfactual by solving a multi-objective optimization problem. Recently, Russell (2019) and Mothilal et al. (2020) focus on finding a set of multiple diverse recourse actions, where the

diversity is imposed by a rule-based approach or by internalize a determinant point process cost in the objective function.

These aforementioned approaches make a fundamental assumption that the machine learning model does not change over time. However, the dire reality suggests that this assumption rarely holds. In fact, data shifts are so common nowadays in machine learning that they have sparkled the emerging field of domain generalization and domain adaptation. Organizations usually retrain models as a response to data shifts and this induces corresponding shifts in the machine learning models parameters, which in turns cause serious concerns for the feasibility of the recourse action in the future (Rawal et al., 2021). In fact, all of the aforementioned approaches design the action which is feasible only with the *current* model parameters, and they provide no feasibility guarantee for the *future* parameters. If a recourse action fails to generate a favorable outcome in the future, then the recourse action may become less beneficial (Venkatasubramanian & Alfano, 2020), the pledge of a brighter outcome is shattered, and the trust on the machine learning system is lost (Rudin, 2019; Ribeiro et al., 2016).

To tackle this challenge, Upadhyay et al. (2021) proposed ROAR, a framework for generating instance level recourses (counterfactual explanations) that are robust to shifts in the underlying predictive model. ROAR used a robust optimization approach that hedges against an uncertainty set containing plausible values of the future model parameters. However, it is well-known that robust optimization solutions can be overly conservative because they may hedge against a pathological parameter in the uncertainty set. A promising approach that can promote robustness, while at the same time prevent from over-conservatism is the distributionally robust optimization framework (El Ghaoui et al., 2003; Delage & Ye, 2010; Rahimian & Mehrotra, 2019; Bertsimas et al., 2018). This framework models the future model parameters as random variables whose underlying distribution is unknown, but is likely to be contained in an ambiguity set. The solution is designed to counter the worst-case distribution in the ambiguity set in a min-max sense. Distributionally robust optimization is also gaining popularity in many estimation and prediction tasks in machine learning (Namkoong & Duchi, 2017; Kuhn et al., 2019).

**Contributions.** This paper combines ideas and techniques from two principal branches of explainable artificial intelligence: counterfactual explanations and robustness, in order to resolve the recourse problem under uncertainty. Concretely, our main contributions are the following:

1. We propose the framework of Distributionally Robust Recourse Action (DiRRAc) for designing a recourse action that is robust to mixture shifts of the model parameters. Our DiRRAc maximizes the probability that the action is feasible with respect to a mixture shift of model parameters, while at the same time cap the action in the neighborhood of the input instance. Moreover, the DiRRAc model also hedges against the misspecification of the nominal distribution using a min-max form with a mixture ambiguity set prescribed by moment information.

2. We reformulate the DiRRAc problem into a finite-dimensional optimization problem with an explicit objective function. We also provide a projected gradient descent to solve the resulting reformulation with convergence guarantees.

3. We extend our DiRRAc framework along several axis to handle mixture weight uncertainty, to minimize the worst-case component probability of receiving unfavorable outcome, and also to inject the Gaussian parametric information.

We first describe the recourse action problem with mixture shift in Section 2. In Section 3, we present our proposed DiRRAc framework, its reformulation and the numerical routine for solving it. The extension to the parametric Gaussian setting will be subsequently discussed in Section 4. Section 5 reports the numerical experiments showing the benefits of the DiRRAc framework and its extensions.

**Notations.** For each integer $K$, we have $[K] = \{1, \ldots, K\}$. We use $\mathbb{S}^d_+$ ($\mathbb{S}^d_{++}$) to denote the space of symmetric positive semidefinite (definite, respectively) matrices. For any $A \in \mathbb{R}^{m \times m}$, the trace operator is defined as $\mathrm{Tr}\left[A\right] = \sum_{i=1}^d A_{ii}$. We write $\mathbb{Q}_k \sim (\mu_k, \Sigma_k)$ to denote that the distribution $\mathbb{Q}_k$ has mean vector $\mu_k$ and covariance matrix $\Sigma_k$. If additionally $\mathbb{Q}_k$ is Gaussian, we write $\mathbb{Q}_k \sim \mathcal{N}(\mu_k, \Sigma_k)$. With a slight abuse of notation, $\mathbb{Q} \sim (\mathbb{Q}_k, p_k)_{k \in [K]}$ means $\mathbb{Q}$ is a mixture of $K$ component distributions, the $k$-th component has weight $p_k$ and distribution $\mathbb{Q}_k$.

## 2 RECOURSE ACTION UNDER MIXTURE SHIFTS

We consider a binary classification setting with label $\mathcal{Y} = \{0, 1\}$, where $0$ represents the unfavorable outcome while $1$ denotes the favorable one. The covariate space is $\mathbb{R}^d$, and any linear classifier $\mathcal{C}_\theta : \mathbb{R}^d \to \mathcal{Y}$ characterized by the $d$-dimensional parameter $\theta$ is of the form

$$\mathcal{C}_\theta(x) = \begin{cases} 1 & \text{if } \theta^\top x \geq 0, \\ 0 & \text{otherwise.} \end{cases}$$

Note that the bias term can be internalized into $\theta$ by adding an extra dimension, and thus it is omitted.

Suppose that at this moment ($t = 0$), the current classifier is parametrized by $\theta_0$, and we are given an input instance $x_0 \in \mathbb{R}^d$ with *un*favorable outcome, that is, $\mathcal{C}_{\theta_0}(x_0) = 0$. One period of time from now ($t = 1$), the parameters of the predictive model will change stochastically, and are represented by a $d$-dimensional random vector $\tilde{\theta}$. This paper focuses on finding a recourse action $x$ which is reasonably close to the instance $x_0$, and at the same time, has a high probability of receiving a favorable outcome in the future. Figure 1 gives a bird's eye view of the setup

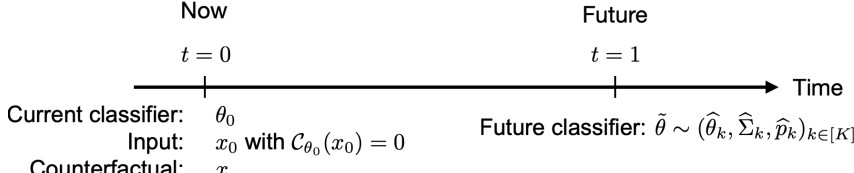

Figure 1: A canonical setup of the recourse action under mixture shifts problem.

To measure the closeness between the action $x$ and the input $x_0$, we assume that the covariate space is endowed with a non-negative, continuous cost function $c$. In addition, suppose temporarily that $\tilde{\theta}$ follows a distribution $\widehat{\mathbb{P}}$. Because maximizing the probability of the favorable outcome is equivalent to minimizing the probability of the unfavorable outcome, the recourse can be found by solving

$$\begin{aligned} \min_x \quad & \widehat{\mathbb{P}}(\mathcal{C}_{\tilde{\theta}}(x) = 0) \\ \text{s.t.} \quad & x \in \mathbb{X}, \ c(x, x_0) \leq \delta. \end{aligned} \tag{1}$$

The parameter $\delta \geq 0$ in (1) governs how far a recourse action can be from the input instance $x_0$. Note that we constrain $x$ in a set $\mathbb{X}$ which captures operational constraints, for example, the highest education of a credit applicant should not be decreasing over time.

In this paper, we model the random vector $\tilde{\theta}$ using a finite mixture of distributions with $K$ components, the mixture weights are $\widehat{p}$ satisfying $\sum_{k \in [K]} \widehat{p}_k = 1$. Each component in the mixture represents one specific type of data shifts: the weights $\widehat{p}$ reflect the proportion of the shift types while the component distribution $\widehat{\mathbb{P}}_k$ representing the (conditional) distribution of the future model parameters in the $k$-th shift. Further information on mixture distributions and their applications in machine learning can be found in Murphy (2012, §3.5).

If each $\widehat{\mathbb{P}}_k$ is a Gaussian distribution $\mathcal{N}(\widehat{\theta}_k, \widehat{\Sigma}_k)$, then $\widehat{\mathbb{P}}$ is a mixture of Gaussian distributions. The objective of problem (1) can be expressed as

$$\widehat{\mathbb{P}}(\mathcal{C}_{\tilde{\theta}}(x) = 0) = \sum_{k \in [K]} \widehat{p}_k \widehat{\mathbb{P}}_k(\mathcal{C}_{\tilde{\theta}}(x) = 0) = \sum_{k \in [K]} \widehat{p}_k \Phi\left(\frac{-x^\top \widehat{\theta}_k}{\sqrt{x^\top \widehat{\Sigma}_k x}}\right),$$

where the first equality follows from the law of conditional probability, and $\Phi$ is the cumulative distribution function of a standard Gaussian distribution. Under the Gaussian assumption, we can solve (1) using a projected gradient descent type of algorithm (Boyd & Vandenberghe, 2004).

**Remark 2.1** (Nonlinear models). *Our analysis focuses on linear classifiers, which is a common setup in the literature (Upadhyay et al., 2021; Ustun et al., 2019; Rawal et al., 2021; Karimi et al., 2020; Wachter et al., 2018; Ribeiro et al., 2016). To extend to nonlinear classifiers, we can follow a similar approach as in Rawal & Lakkaraju (2020) and Upadhyay et al. (2021) by first using LIME (Ribeiro et al., 2016) to approximate the nonlinear classifiers locally with an interpretable linear model, then subsequently applying our framework.*

# 3 DISTRIBUTIONALLY ROBUST RECOURSE ACTION FRAMEWORK

Our Distributionally Robust Recourse Action (DiRRAc) framework robustifies formulation (1) by relaxing the parametric assumption and hedging against distribution misspecification. First, we assume that the mixture components $\widehat{\mathbb{P}}_k$ are specified only through moment information, and no particular parametric form of the distribution is imposed. In effect, $\widehat{\mathbb{P}}_k$ is assumed to have mean vector $\widehat{\theta}_k \in \mathbb{R}^d$ and positive definite covariance matrix $\widehat{\Sigma}_k \succ 0$. Second, we leverage ideas from distributionally robust optimization to propose a min-max formulation of (1), in which we consider an *ambiguity set* which contains a family of probability distributions that are sufficiently close to the nominal distribution $\widehat{\mathbb{P}}$. To prescribe the ambiguity set, we use the Gelbrich distance.

**Definition 3.1** (Gelbrich distance). *The Gelbrich distance $\mathbb{G}$ between two tuples $(\theta, \Sigma) \in \mathbb{R}^d \times \mathbb{S}^d_+$ and $(\widehat{\theta}, \widehat{\Sigma}) \in \mathbb{R}^d \times \mathbb{S}^d_+$ amounts to $\mathbb{G}((\theta, \Sigma), (\widehat{\theta}, \widehat{\Sigma})) \triangleq \sqrt{\|\theta - \widehat{\theta}\|_2^2 + \mathrm{Tr}\left[\Sigma + \widehat{\Sigma} - 2(\widehat{\Sigma}^{\frac{1}{2}} \Sigma \widehat{\Sigma}^{\frac{1}{2}})^{\frac{1}{2}}\right]}$.*

It is easy to verify that $\mathbb{G}$ is non-negative, symmetric and it vanishes to zero if and only if $(\theta, \Sigma) = (\widehat{\theta}, \widehat{\Sigma})$. Further, $\mathbb{G}$ is a distance on $\mathbb{R}^d \times \mathbb{S}^d_+$ because it coincides with the type-2 Wasserstein distance between two Gaussian distributions $\mathcal{N}(\mu, \Sigma)$ and $\mathcal{N}(\widehat{\mu}, \widehat{\Sigma})$ (Givens & Shortt, 1984). Distributionally robust formulations with moment information prescribed by the $\mathbb{G}$ distance are computationally tractable under mild conditions, deliver reasonable performance guarantees and also generate a conservative approximation of the Wasserstein distributionally robust optimization problem (Kuhn et al., 2019).

In this paper, we use the Gelbrich distance $\mathbb{G}$ to form a neighborhood around each $\widehat{\mathbb{P}}_k$ as

$$\mathcal{B}_k(\widehat{\mathbb{P}}_k) \triangleq \left\{ \mathbb{Q}_k : \mathbb{Q}_k \sim (\theta_k, \Sigma_k), \ \mathbb{G}((\theta_k, \Sigma_k), (\widehat{\theta}_k, \widehat{\Sigma}_k)) \leq \rho_k \right\}.$$

Intuitively, one can view $\mathcal{B}_k(\widehat{\mathbb{P}}_k)$ as a ball centered at the nominal component $\widehat{\mathbb{P}}_k$ of radius $\rho_k \geq 0$ prescribed using the distance $\mathbb{G}$. This component set $\mathcal{B}_k(\widehat{\mathbb{P}}_k)$ is non-parametric, and the first two moments of $\mathbb{Q}_k$ are sufficient to decide whether $\mathbb{Q}_k$ belongs to $\mathcal{B}_k(\widehat{\mathbb{P}}_k)$. Moreover, if $\mathbb{Q}_k \in \mathcal{B}_k(\widehat{\mathbb{P}}_k)$, then any distribution $\mathbb{Q}'_k$ with the same mean vector and covariance matrix as $\mathbb{Q}_k$ also belongs to $\mathcal{B}_k(\widehat{\mathbb{P}}_k)$. Notice that even when the radius $\rho_k$ is zero, the component set $\mathcal{B}_k(\widehat{\mathbb{P}}_k)$ does not collapse into a singleton. Instead, if $\rho_k = 0$ then $\mathcal{B}_k(\widehat{\mathbb{P}}_k)$ still contains *all* distributions of the same moment $(\widehat{\theta}_k, \widehat{\Sigma}_k)$ with the nominal component distribution $\widehat{\mathbb{P}}_k$, and consequentially it possesses the robustification effects against the parametric assumption on $\widehat{\mathbb{P}}_k$. The component sets are utilized to construct the ambiguity set for the mixture distribution as

$$\mathcal{B}(\widehat{\mathbb{P}}) \triangleq \left\{ \mathbb{Q} : \exists \mathbb{Q}_k \in \mathcal{B}_k(\widehat{\mathbb{P}}_k) \ \forall k \in [K] \text{ such that } \mathbb{Q} \sim (\mathbb{Q}_k, \widehat{p}_k)_{k \in [K]} \right\}.$$

Any $\mathbb{Q} \in \mathcal{B}(\widehat{\mathbb{P}})$ is also a mixture distribution with $K$ components, with the same mixture weights $\widehat{p}$. Thus, $\mathcal{B}(\widehat{\mathbb{P}})$ contains all perturbations of $\widehat{\mathbb{P}}$ induced separately on each component by $\mathcal{B}_k(\widehat{\mathbb{P}}_k)$.

We are now ready to introduce our DiRRAc model, which is a min-max problem of the form

$$\begin{aligned} \inf_{x \in \mathbb{X}} \quad & \sup_{\mathbb{Q} \in \mathcal{B}(\widehat{\mathbb{P}})} \mathbb{Q}(\mathcal{C}_{\tilde{\theta}}(x) = 0) \\ \text{s.t.} \quad & c(x, x_0) \leq \delta \\ & \sup_{\mathbb{Q}_k \in \mathcal{B}_k(\widehat{\mathbb{P}}_k)} \mathbb{Q}_k(\mathcal{C}_{\tilde{\theta}}(x) = 0) < 1 \qquad \forall k \in [K]. \end{aligned} \qquad (2)$$

The objective of (2) is to minimize the worst-case probability of unfavorable outcome of the recourse action. Moreover, the last constraint imposes that for each component, the worst-case conditional probability of unfavorable outcome should be strictly less than 1. Put differently, this last constraint requires that the action should be able to lead to favorable outcome for *any* distribution in $\mathcal{B}_k(\widehat{\mathbb{P}}_k)$. By definition, each supremum subproblem in (2) is an infinite-dimensional maximization problem over the space of probability distributions, and thus it is inherently difficult. Fortunately, because we use the Gelbrich distance to prescribe the set $\mathcal{B}_k(\widehat{\mathbb{P}}_k)$, we can solve these maximization problems

analytically. This consequentially leads to a closed-form reformulation of the DiRRAc model into a finite-dimensional problem. Next, we will reformulate the DiRRAc problem (2), provide a sketch of the proof and propose a numerical solution routine.

## 3.1 REFORMULATION OF DiRRAc

Each supremum in (2) is an infinite-dimensional optimization problem on the space of probability distributions. We now show that (2) can be reformulated as a finite-dimensional problem. Towards this end, let $\mathcal{X}$ be the following $d$-dimensional set

$$\mathcal{X} \triangleq \left\{ x \in \mathbb{X} : \ c(x, x_0) \leq \delta, \quad -\widehat{\theta}_k^\top x + \rho_k \|x\|_2 < 0 \quad \forall k \in [K] \ \right\}. \tag{3}$$

The next theorem asserts that the DiRRAc problem (2) can be reformulated as a $d$-dimensional optimization problem with an explicit, but complicated, objective function.

**Theorem 3.2** (Equivalent form of DiRRAc). *Problem (2) is equivalent to the following finite-dimensional problem*

$$\inf_{x \in \mathcal{X}} \ \sum_{k \in [K]} \widehat{p}_k \left( \frac{\rho_k \widehat{\theta}_k^\top x \|x\|_2 + \sqrt{x^\top \widehat{\Sigma}_k x} \sqrt{(\widehat{\theta}_k^\top x)^2 + x^\top \widehat{\Sigma}_k x - \rho_k^2 \|x\|_2^2}}{(\widehat{\theta}_k^\top x)^2 + x^\top \widehat{\Sigma}_k x} \right)^2. \tag{4}$$

## 3.2 PROOF SKETCH

We now sketch the proof of Theorem 3.2. For any component $k \in [K]$, define the following worst-case probability of unfavorable outcome function

$$f_k(x) \triangleq \sup_{\mathbb{Q}_k \in \mathcal{B}_k(\widehat{\mathbb{P}}_k)} \mathbb{Q}_k(\mathcal{C}_{\tilde{\theta}}(x) = 0) = \sup_{\mathbb{Q}_k \in \mathcal{B}_k(\widehat{\mathbb{P}}_k)} \mathbb{Q}_k(\tilde{\theta}^\top x \leq 0) \qquad \forall k \in [K]. \tag{5}$$

To proceed, we rely on the following elementary result from Nguyen (2019, Lemma 3.31).

**Lemma 3.3** (Worst-case Value-at-Risk). *For any $x \in \mathbb{R}^d$ and $\beta \in (0, 1)$, we have*

$$\inf \left\{ \tau : \sup_{\mathbb{Q}_k \in \mathcal{B}_k(\widehat{\mathbb{P}}_k)} \mathbb{Q}_k(\tilde{\theta}^\top x \leq -\tau) \leq \beta \right\} = -\widehat{\theta}_k^\top x + \sqrt{\frac{1-\beta}{\beta}} \sqrt{x^\top \widehat{\Sigma}_k x} + \frac{\rho_k}{\sqrt{\beta}} \|x\|_2. \tag{6}$$

Note that the left-hand side of (6) is the worst-case Value-at-Risk with respect to the ambiguity set $\mathcal{B}_k(\widehat{\mathbb{P}}_k)$. Leveraging this result, the next proposition provides the analytical form of $f_k(x)$.

**Proposition 3.4** (Worst-case probability). *For any $k \in [K]$ and $(\widehat{\theta}_k, \widehat{\Sigma}_k, \rho_k) \in \mathbb{R}^d \times \mathbb{S}_+^d \times \mathbb{R}_+$, define the following constants $A_k \triangleq -\widehat{\theta}_k^\top x$, $B_k \triangleq \sqrt{x^\top \widehat{\Sigma}_k x}$, and $C_k \triangleq \rho_k \|x\|_2$. We have*

$$f_k(x) \triangleq \sup_{\mathbb{Q}_k \in \mathcal{B}_k(\widehat{\mathbb{P}}_k)} \mathbb{Q}_k(\tilde{\theta}^\top x \leq 0) = \begin{cases} 1 & \text{if } A_k + C_k \geq 0, \\ \left( \frac{-A_k C_k + B_k \sqrt{A_k^2 + B_k^2 - C_k^2}}{A_k^2 + B_k^2} \right)^2 \in (0, 1) & \text{if } A_k + C_k < 0. \end{cases}$$

The proof of Theorem 3.2 follows by noticing that the DiRRAc problem (2) can be reformulated using the elementary functions $f_k$ as

$$\min_{x \in \mathbb{X}} \left\{ \sum_{k \in [K]} \widehat{p}_k f_k(x) \ : \ c(x, x_0) \leq \delta, \ f_k(x) \leq 0 \quad \forall k \in [K] \right\},$$

where the objective function follows from the definition of the set $\mathcal{B}(\widehat{\mathbb{P}})$. It suffices now to combine with Proposition 3.4 to obtain the necessary result. The detailed proof is relegated to the Appendix.

### 3.3 PROJECTED GRADIENT DESCENT ALGORITHM

We consider in this section an iterative numerical routine to solve the DiRRAc problem in the equivalent form (4). First, notice that the second constraint that defines $\mathcal{X}$ in (3) is a strict inequality, thus the set $\mathcal{X}$ is open. We thus modify slightly this constraint by considering the following set

$$\mathcal{X}_\varepsilon = \left\{ x \in \mathbb{X} \; : \; c(x, x_0) \leq \delta, \; -\widehat{\theta}_k^\top x + \rho_k \|x\|_2 \leq -\varepsilon \quad \forall k \in [K] \right\}$$

for some value $\varepsilon > 0$ sufficiently small. Moreover, if the parameter $\delta$ is too small, it may happen that the feasible set $\mathcal{X}_\varepsilon$ becomes empty. Let $\delta_{\min} \in \mathbb{R}_+$ be defined as the optimal value of the following optimization problem

$$\delta_{\min} \triangleq \begin{cases} \inf & c(x, x_0) \\ \text{s.t.} & x \in \mathbb{X}, \; -\widehat{\theta}_k^\top x + \rho_k \|x\|_2 \leq -\varepsilon \qquad \forall k \in [K]. \end{cases} \tag{7}$$

Then it is easy to see that $\mathcal{X}_\varepsilon$ is non-empty whenever $\delta \geq \delta_{\min}$. In addition, because $c$ is continuous and $\mathbb{X}$ is closed, the set $\mathcal{X}_\varepsilon$ is compact. In this case, we can consider problem (4) with the feasible set being $\mathcal{X}_\varepsilon$, for which the optimal solution is guaranteed to exist.

Let us now define the projection operator $\mathrm{Proj}_{\mathcal{X}_\varepsilon}$ as

$$\mathrm{Proj}_{\mathcal{X}_\varepsilon}(x') = \arg\min \left\{ \|x - x'\|_2^2 \; : \; x \in \mathcal{X}_\varepsilon \right\}.$$

If $\mathbb{X}$ is convex and $c(\,\cdot\,, x_0)$ is a convex function, then $\mathcal{X}_\varepsilon$ is also convex, and the projection operation can be efficiently computed using convex optimization. In particular, suppose that $c(x, x_0) = \|x - x_0\|_2$ is the Euclidean norm and $\mathbb{X}$ is second-order cone representable, then the projection is equivalent to a second-order cone program, and can be solved using off-the-shelf solvers such as GUROBI or Mosek (MOSEK ApS, 2019). The projection operator $\mathrm{Proj}_{\mathcal{X}_\varepsilon}$ now forms the building block of a projected gradient descent algorithm with a backtracking linesearch. The details regarding the algorithm, along with the convergence guarantee, are presented in Appendix E.

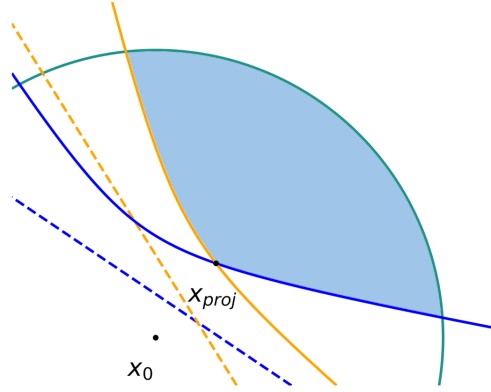

Figure 2: Shaded area represents $\mathcal{X}$. Circular arc represents the proximity constraint $c(x, x_0) = \delta$. Dashed lines represent the hyperplane $-\widehat{\theta}_k^\top x = 0$, elliptic curves represent the robust margin $-\widehat{\theta}_k^\top x + \rho_k \|x\| = 0$. Increasing $\rho_k$ brings the elliptic curves farther away from the dash lines, and the set $\mathcal{X}$ moves deeper inside the favorable prediction region.

## 4 GAUSSIAN DiRRAc FRAMEWORK

We here revisit the Gaussian assumption on the component distributions, and propose the parametric Gaussian DiRRAc framework. We make the temporary assumption that $\widehat{\mathbb{P}}_k$ are Gaussian for all $k \in [K]$, and we will robustify against only the misspecification of the nominal mean vector and covariance matrix $(\widehat{\theta}_k, \widehat{\Sigma}_k)$. To do this, we first construct the Gaussian component ambiguity sets

$$\forall k : \quad \mathcal{B}_k^\mathcal{N}(\widehat{\mathbb{P}}_k) \triangleq \left\{ \mathbb{Q}_k : \mathbb{Q}_k \sim \mathcal{N}(\theta_k, \Sigma_k), \; \mathbb{G}((\theta_k, \Sigma_k), (\widehat{\theta}_k, \widehat{\Sigma}_k)) \leq \rho_k \right\},$$

where the superscript emphasizes that the ambiguity sets are neighborhoods in the space of Gaussian distributions. The resulting ambiguity set for the mixture distribution is

$$\mathcal{B}^\mathcal{N}(\widehat{\mathbb{P}}) = \left\{ \mathbb{Q} \; : \; \exists \mathbb{Q}_k \in \mathcal{B}_k^\mathcal{N}(\widehat{\mathbb{P}}_k) \; \forall k \in [K] \text{ such that } \mathbb{Q} \sim (\mathbb{Q}_k, \widehat{p}_k)_{k \in [K]} \right\}.$$

The Gaussian DiRRAc problem is formally defined as

$$\begin{aligned} \min_{x \in \mathbb{X}} \quad & \sup_{\mathbb{Q} \in \mathcal{B}^\mathcal{N}(\widehat{\mathbb{P}})} \mathbb{Q}(\mathcal{C}_{\tilde{\theta}}(x) = 0) \\ \text{s.t.} \quad & c(x, x_0) \leq \delta \\ & \sup_{\mathbb{Q}_k \in \mathcal{B}_k^\mathcal{N}(\widehat{\mathbb{P}}_k)} \mathbb{Q}_k(\mathcal{C}_{\tilde{\theta}}(x) = 0) < \tfrac{1}{2} \qquad \forall k \in [K]. \end{aligned} \tag{8}$$

Similar to Section 3, we will provide the reformulation of the Gaussian DiRRAc formulation and a sketch of the proof in the sequence. Note that the last constraint in (8) has margin $\frac{1}{2}$ instead of 1 as in the DiRRAc problem (2). The detailed reason is revealed in the proof sketch in Section 4.2.

## 4.1 Reformulation of Gaussian DiRRAc

Remind that the feasible set $\mathcal{X}$ is defined as in equation 3. The next theorem asserts the equivalent form of the Gaussian DiRRAc problem (8).

**Theorem 4.1** (Gaussian DiRRAc reformulation)**.** *The Gaussian DiRRAc problem (8) is equivalent to the following optimization problem*

$$\min_{x \in \mathcal{X}} 1 - \sum_{k \in [K]} \widehat{p}_k \Phi\left( \frac{(\widehat{\theta}_k^\top x)^2 - \rho_k^2 \|x\|_2^2}{\widehat{\theta}_k^\top x \sqrt{x^\top \widehat{\Sigma}_k x} + \rho_k \|x\|_2 \sqrt{(\widehat{\theta}_k^\top x)^2 + x^\top \widehat{\Sigma}_k x - \rho_k^2 \|x\|_2^2}} \right). \tag{9}$$

Problem (9) can be solved using the projected gradient descent algorithm discussed in Section 3.3. Note that the gradient of the objective function can be evaluated easily using the chain rule.

## 4.2 Proof Sketch

The proof of Theorem 4.1 relies on the following result which asserts the analytical form of the worst-case Value-at-Risk under parametric Gaussian ambiguity set (Nguyen, 2019, Lemma 3.31).

**Lemma 4.2** (Worst-case Gaussian Value-at-Risk)**.** *For any $x \in \mathbb{R}^d$ and $\beta \in (0, \frac{1}{2}]$, we have*

$$\inf \left\{ \tau : \sup_{\mathbb{Q}_k \in \mathcal{B}_k^{\mathcal{N}}(\widehat{\mathbb{P}}_k)} \mathbb{Q}_k(\tilde{\theta}^\top x \le -\tau) \le \beta \right\} = -\widehat{\theta}_k^\top x + t \sqrt{x^\top \widehat{\Sigma}_k x} + \rho \sqrt{1 + t^2} \|x\|_2 \tag{10}$$

*with $t = \Phi^{-1}(1 - \beta)$.*

It is important to note that Lemma 4.2 is only valid for $\beta \in (0, 0.5]$. Indeed, for $\beta > \frac{1}{2}$, evaluating the infimum problem in the left-hand side of (10) requires solving a *non-convex* optimization problem as $t = \Phi^{-1}(1 - \beta) < 0$. As a consequence, the last constraint of the Gaussian DiRRAc formulation (8) is capped at a probability value of 0.5 to ensure the convexity of the feasible set in the reformulation (9). The proof of Theorem 4.1 follows a similar line of argument as for the DiRRAc formulation, the details are relegated to the appendix.

## 5 Numerical Experiments

In this section, we evaluate the performance of our DiRRAc framework on popular benchmarks. We will compare our proposed DiRRAc model (2) and Gaussian DiRRAc model (8) against three state-of-the-art methods: the Robust and Reliable Algorithmic Recourse (ROAR) (Upadhyay et al., 2021), Actionable Recourse (AR) in linear classification (Ustun et al., 2019) and Model Agnostic Contrastive Explanations (MACE) (Karimi et al., 2020). Throughout, we use the $l_1$ distance $c(x, x_0) = \|x - x_0\|_1$. Complementary results and details about the datasets and the experiment setup are provided in Appendix A. All codes and results can be accessed from https://anonymous.4open.science/r/DiRRAc.

**Results on synthetic data.** We synthesize 2-dimensional data by using $K = 3$ different shifts similar to Upadhyay et al. (2021): mean shift, covariance shift, and both shifts. First, we fix the unshifted conditional distributions with $X|Y = y \sim \mathcal{N}(\mu_y, \Sigma_y) \; \forall y \in \mathcal{Y}$. For mean shift, we replace $\mu_0$ by $\mu_0^{\text{shift}} = \mu_0 + [\alpha, 0]^\top$, where $\alpha$ is a mean shift magnitude. For covariance shift, we replace $\Sigma_0$ by $\Sigma_0^{\text{shift}} = (1+\beta)\Sigma_0$, where $\beta$ is a covariance shift magnitude. For mean and covariance shift, we replace $(\mu_0, \Sigma_0)$ by $(\mu_0^{\text{shift}}, \Sigma_0^{\text{shift}})$. We generate 500 samples each class from the unshifted distribution with $\mu_0 = [-3; -3]$, $\mu_1 = [3; 3]$, and $\Sigma_0 = \Sigma_1 = I$. To estimate $\widehat{\theta}_k$ and $\widehat{\Sigma}_k$ for synthetic data, we define valid mixture weights $\widehat{p}$, generate data for each component for 100 times with the same ratio as the mixture weight. We train 100 logistic classifiers to compute the empirical mean $\widehat{\theta}_k$ and the empirical covariance matrix $\widehat{\Sigma}_k$ for the $k$-th component. We generate recourse for

each test instance that belongs to negative class. Finally, we compute the empirical validity as the fraction of instances that are still valid with respect to the shifted classifiers. The results in Figure 3 demonstrate that recourses generated by our framework are robust to model shifts, other baselines have low validity with even a small shift magnitude.

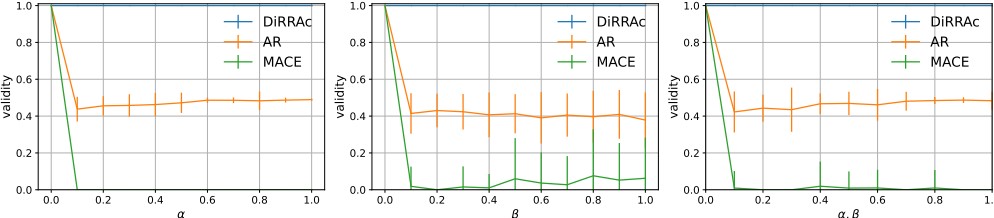

Figure 3: Impact of magnitude of distribution shifts to empirical validity

**Real-world data.** We use three real-world datasets which capture different data distribution shifts (Dua & Graff, 2017): (i) the German credit dataset, which captures a correction shift. (ii) the Small Business Administration (SBA) dataset, which captures a temporal shift. (iii) the Student performance dataset, which captures a geospatial shift. Each dataset contains original data and shifted data. We normalize all continuous features to $[0, 1]$. Similar to Mothilal et al. (2020), we use one-hot encodings for categorial features, then consider them as continuous features in $[0, 1]$. To ease the comparison, we choose $K = 1$. To estimate $(\widehat{\theta}_1, \widehat{\Sigma}_1)$, we split randomly 80% of the original dataset and train a logistic classifier. This process is repeated independently 100 times to obtain 100 observations of the model parameters, then we compute the empirical mean and covariance matrix for $(\widehat{\theta}_1, \widehat{\Sigma}_1)$. In parallel, we randomly split 80-20 the shifted dataset 100 times, and each time train a logistic classifier on the training set. This procedure generates 100 future model parameters.

To measure the performance of each method, we do a (80% training, 20% testing) split of the original dataset, train a linear classifier on the training data. and generate recourse for each test instance that is classified as unfavorable. To compute $M_1$ validity, we split randomly 80% of the original data 100 times and train 100 logistic classifier. The $M_1$ validity is computed by the empirical validity on those 100 model parameters. The $M_2$ validity is measured by the empirical validity on the 100 future model parameters, and we also compute the $l_1$ and $l_2$ distance between the recourse and the original instance. The results in Table 1 demonstrate that our DiRRAc have high validity, while keeping the $l_1$ and $l_2$ cost low. ROAR has high validity in all three datasets, but also has higher cost than our framework.

Table 1: Benchmark of $M_1$ validity, $M_2$ validity, $l_1$ and $l_2$ cost on different real-world datasets.

| Dataset | Methods | $M_1$ validity | $M_2$ validity | $l_1$ cost | $l_2$ cost |
|---|---|---|---|---|---|
| German Credit | AR | $0.73 \pm 0.25$ | $0.78 \pm 0.00$ | $1.26 \pm 0.68$ | $0.94 \pm 0.41$ |
| | MACE | $0.87 \pm 0.15$ | $0.97 \pm 0.00$ | $2.11 \pm 0.86$ | $1.20 \pm 0.47$ |
| | ROAR | $1.00 \pm 0.00$ | $1.00 \pm 0.00$ | $2.60 \pm 0.40$ | $1.08 \pm 0.16$ |
| | DiRRAc | $\mathbf{1.00} \pm 0.00$ | $\mathbf{1.00} \pm 0.00$ | $2.09 \pm 0.43$ | $0.96 \pm 0.18$ |
| | Gaussian DiRRAc | $1.00 \pm 0.00$ | $0.93 \pm 0.05$ | $\mathbf{0.73} \pm 0.47$ | $\mathbf{0.47} \pm 0.38$ |
| SBA | AR | $0.26 \pm 0.24$ | $0.42 \pm 0.14$ | $3.41 \pm 2.10$ | $1.56 \pm 0.76$ |
| | MACE | $1.00 \pm 0.00$ | $1.00 \pm 0.00$ | $6.85 \pm 0.56$ | $2.50 \pm 0.11$ |
| | ROAR | $1.00 \pm 1.00$ | $1.00 \pm 0.00$ | $2.25 \pm 0.55$ | $0.98 \pm 0.23$ |
| | DiRRAc | $\mathbf{1.00} \pm 0.00$ | $\mathbf{1.00} \pm 0.00$ | $\mathbf{1.13} \pm 0.43$ | $\mathbf{0.82} \pm 0.31$ |
| | Gaussian DiRRAc | $1.00 \pm 0.01$ | $1.00 \pm 0.00$ | $1.14 \pm 0.42$ | $0.83 \pm 0.30$ |
| Student Performance | AR | $0.28 \pm 0.08$ | $0.35 \pm 0.12$ | $1.18 \pm 0.99$ | $0.82 \pm 0.60$ |
| | MACE | $0.66 \pm 0.12$ | $0.57 \pm 0.10$ | $0.81 \pm 0.40$ | $\mathbf{0.51} \pm 0.24$ |
| | ROAR | $1.00 \pm 0.01$ | $0.98 \pm 0.02$ | $1.70 \pm 0.27$ | $0.81 \pm 0.13$ |
| | DiRRAc | $\mathbf{1.00} \pm 0.00$ | $\mathbf{0.99} \pm 0.02$ | $\mathbf{0.74} \pm 0.18$ | $0.63 \pm 0.14$ |
| | Gaussian DiRRAc | $1.00 \pm 0.00$ | $0.98 \pm 0.02$ | $0.74 \pm 0.18$ | $0.74 \pm 0.18$ |

**Nonlinear models.** Following the previous work as in Rawal et al. (2021) and Upadhyay et al. (2021), we adapt our framework and other baselines to non-linear models by first generating local

linear approximations using LIME (Ribeiro et al., 2016). LIME is a popular method that explains the predictions of a machine learning model by learning an interpretable model locally around an input instance $x_0$. To obtain a local explanation, LIME synthesizes perturbed examples in the local neighborhood of $x_0$, queries the predictions of models for these examples, and then trains an interpretable model based on these labeled synthetic examples.

For each instance $x_0$, we first do a 80-20 split on the original dataset and train a MLPs classifier on the training data. Then generate a local linear model for MLPs classifier 10 times using LIME with 1000 perturbed samples. To estimate $(\widehat{\theta}_1, \widehat{\Sigma}_1)$, we compute the mean and covariance matrix of parameters $\theta_{x_0}$ of 10 local linear models. We generate recourse for each instance of test data that belongs to negative class . We choose randomly 10% of shifted dataset and concatenate with training data of original dataset, then train a MLPs classifier. This process is repeated 10 times to obtain 10 shifted classifiers. We evaluate the $M_1$ and $M_2$ validity of each method by computing validity of recourses on the original MLPs classifier and the shifted classifiers.

The results in Table 2 demonstrate that our DiRRAc have a higher validity than other baselines in the original and shifted MLPs classifier, while keeping the $l_1$ and $l_2$ cost low.

Table 2: Benchmark of $M_1$ validity, $M_2$ validity, $l_1$ and $l_2$ cost for non-linear models on different real-world datasets.

| Dataset | Methods | $M_1$ validity | $M_2$ validity | $l_1$ cost | $l_2$ cost |
|---|---|---|---|---|---|
| German Credit | AR | $0.67 \pm 0.47$ | $0.59 \pm 0.38$ | $\textbf{1.00} \pm 0.00$ | $1.00$ |
| | MACE | $0.67 \pm 0.47$ | $0.31 \pm 0.22$ | $1.99 \pm 0.29$ | $1.19 \pm 0.13$ |
| | ROAR | $0.87 \pm 0.26$ | $0.66 \pm 0.33$ | $2.66 \pm 0.16$ | $1.21 \pm 0.01$ |
| | DiRRAc | $\textbf{1.00} \pm 0.00$ | $\textbf{0.80} \pm 0.18$ | $1.07 \pm 0.01$ | $1.00 \pm 0.01$ |
| | Gaussian DiRRAc | $0.91 \pm 0.29$ | $0.73 \pm 0.29$ | $1.05 \pm 0.07$ | $\textbf{0.95} \pm 0.17$ |
| SBA | AR | $1.00 \pm 0.00$ | $0.72 \pm 0.27$ | $\textbf{1.02} \pm 0.04$ | $1.00 \pm 0.00$ |
| | MACE | $1.00 \pm 0.00$ | $0.75 \pm 0.16$ | $5.90 \pm 0.45$ | $2.31 \pm 0.07$ |
| | ROAR | $0.91 \pm 0.29$ | $0.91 \pm 0.29$ | $3.34 \pm 0.22$ | $1.07 \pm 0.07$ |
| | DiRRAc | $\textbf{1.00} \pm 0.00$ | $\textbf{0.97} \pm 0.09$ | $1.07 \pm 0.03$ | $\textbf{0.73} \pm 0.11$ |
| | Gaussian DiRRAc | $1.00 \pm 0.00$ | $0.85 \pm 0.23$ | $1.07 \pm 0.03$ | $0.81 \pm 0.09$ |
| Student Performance | AR | $0.47 \pm 0.50$ | $0.41 \pm 0.44$ | $1.02 \pm 0.02$ | $1.02 \pm 0.02$ |
| | MACE | $0.60 \pm 0.49$ | $0.60 \pm 0.49$ | $3.04 \pm 1.61$ | $1.44 \pm 0.57$ |
| | ROAR | $0.95 \pm 0.22$ | $0.86 \pm 0.29$ | $3.67 \pm 0.60$ | $1.24 \pm 0.17$ |
| | DiRRAc | $\textbf{1.00} \pm 0.00$ | $0.94 \pm 0.17$ | $\textbf{0.95} \pm 0.56$ | $\textbf{0.84} \pm 0.26$ |
| | Gaussian DiRRAc | $1.00 \pm 0.00$ | $\textbf{0.96} \pm 0.16$ | $0.95 \pm 0.56$ | $0.87 \pm 0.30$ |

## 6 CONCLUDING REMARKS

In this work, we proposed the Distributionally Robust Recourse Action (DiRRAc) framework to address the problem of recourse robustness to model shifts. We introduced a distributionally robust optimization approach for generating robust recourse using a projected gradient descent algorithm. Furthermore, we also discuss several extensions of our framework. The experiments with synthetic and real-world datasets demonstrated that our framework has the ability to generate recourse that are robust to model shifts under different types of data distribution shifts. We also showed that our framework can be adapted to different model types, linear and non-linear models.

**Remark 6.1** (Extensions). *The DiRRAc framework can be extended to hedge against the misspecification of the mixture weights $\widehat{p}$. Alternatively, the objective function of DiRRAc can be modified to minimize the worst-case component probability. These extensions are explored in Section C. Corresponding extensions for the Gaussian DiRRAc framework are presented in Section D.*

**Remark 6.2** (Choice of ambiguity set). *The distributionally robust result in this paper relies fundamentally on the design of the ambiguity sets using a Gelbrich distance on the moment space. This Gelbrich ambiguity set leads to the $\|\cdot\|_2$-regularizations of the worst-case Value-at-Risk in Lemmas 3.3 and 4.2. If we consider other moment ambiguity sets, for example, the moment bounds in Delage & Ye (2010) or the Kullback-Leibler-type sets in Taskesen et al. (2021), then these regularization equivalence are not available, and there is no trivial way to extend the distributionally robust results to provide the reformulation of the (Gaussian) DiRRAc framework.*

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

# A  ADDITIONAL EXPERIMENT RESULTS

**Synthetic data.** For synthetic data, we perform experiments on datasets capturing 3 types of data distribution shift: mean shift, covariance shift, mean and covariance shift (both shift)

We define the adaptive mean and covariance shift magnitude as $\alpha = \mu_{adapt} \times iter$, $\beta = \Sigma_{adapt} * iter$ with $\mu_{adapt}, \Sigma_{adapt}$ are the factor of data shifts, $iter$ is the index of iterative loop of synthesizing process.

To visualize feasible set as Figure 2, we generate synthetic data with the following parameters: $\mu_0 = [-3; -3]$, $\mu_1 = [3; 3]$, and $\Sigma_0 = \Sigma_1 = I$. Then, we split data to training set and test set (80% for training and 20% for testing), train the original classifier on training set and then choose one instance on test set that is classified as negative class to visualize the feasible set. For data distribution shifts, we generate mean shifts and covariance shifts 50 times each type with adaptive mean and covariance shift magnitude, with the parameters $\mu_{adapt} = \Sigma_{adapt} = 0.1$. Then we train and get the parameters $\widehat{\theta}_k$ and $\widehat{\widehat{\Sigma}}_k$ same as above.

Table 3: Parameters for the feasible set visualization experiment in Figure 2

| Parameters | Values |
|---|---|
| $K$ | 2 |
| $\delta_{\mathrm{add}}$ | 2 |
| $\widehat{p}$ | $[0.5, 0.5]$ |
| $\rho$ | $[1, 1]$ |
| $\lambda$ | 0.7 |
| $\zeta$ | 1 |

Table 4: Parameters for the impact of data distribution shifts experiment in Figure 3

| Parameters | Values |
|---|---|
| $K$ | 3 |
| $\delta_{\mathrm{add}}$ | 0.2 |
| $\widehat{p}$ | $[0.3, 0.4, 0.3]$ |
| $\rho$ | $[0, 0, 0]$ |
| $\lambda$ | 0.7 |
| $\zeta$ | 1 |

To visualize the data and decision boundaries of linear classifiers, we generate synthetic data with the same parameters as above. Then we train 4 classifiers on original data and synthetic capturing 3 types of distribution shifts and visualize the decision boundaries as Figure 4. Then we synthesize shifted data 100 times, 33 mean shifts, 33 covariance shifts and 34 mean and covariance shifts (both shifts) and visualize 100 model's parameters in 2D as Figure 5.

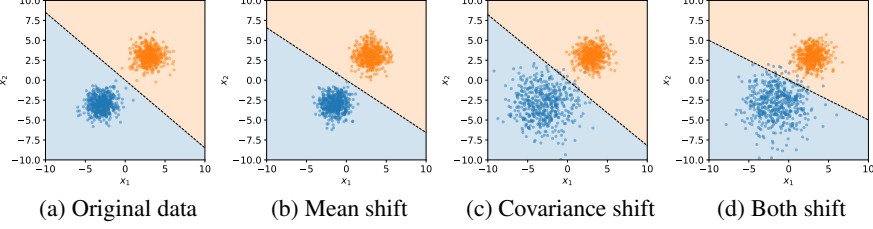

(a) Original data      (b) Mean shift      (c) Covariance shift      (d) Both shift

Figure 4: Synthetic data shifts and the corresponding model parameter shifts (decision boundaries).

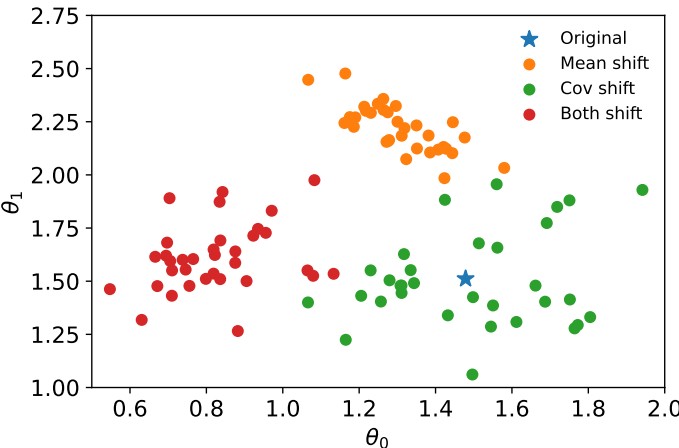

Figure 5: Parameter $\theta$ of the classifier with different types of data distribution shifts

To evaluate how magnitude of upper bound cost affects the validity of our models, we use different values of upper bound cost. We define $\delta = \delta_{\min} + \delta_{\mathrm{add}}$ with $\delta_{\min}$ is defined in (7). We use the same parameters as above for generating synthetic data. For each value of $\delta_{\mathrm{add}}$, we evaluate the validity by using 100 classifiers trained on 100 different data distribution shifts with 3 different shift types.

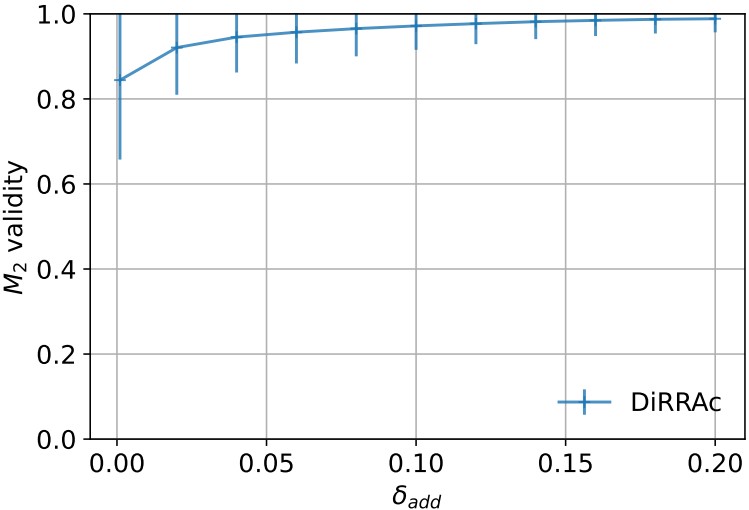

Figure 6: Impact of choosing $\delta_{\mathrm{add}}$ to the validity of DiRRAc

Additional results for the Impact of magnitude of distribution shifts to empirical validity.

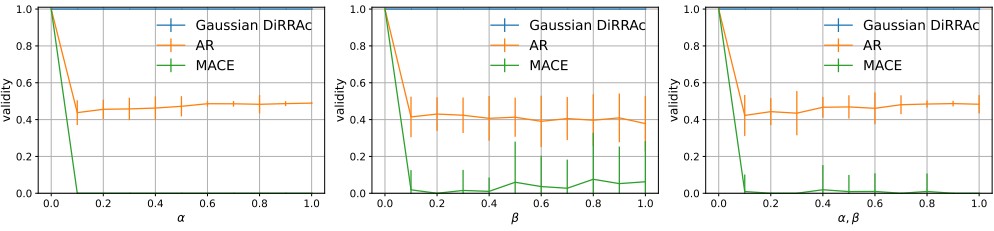

Figure 7: Impact of magnitude of distribution shifts to empirical validity of Gaussian DiRRAc

To evaluate the trade-offs between the $l_1$ cost and the validity of DiRRAc, we benchmark these two criteria by running DiRRAc with different values of $\delta$. We define $\delta = \delta_{\min} + \delta_{\mathrm{add}}$ as before. For each value of $\delta$, we evaluate the $l_1$ cost and validity by using 100 classifiers trained on 100 different data distribution shifts with 3 different shift types.

We also evaluate the trade-offs between $l_1$ cost and validity of DiRRAc using 20 instances. In the comparison with ROAR, we evaluate $l_1$ cost and validity of DiRRAc and ROAR using 10 instances. We use a wide range of parameters for ROAR and compute the $l_1$ cost and validity of recourses. We report the results of 2 experiments in Figure 8 and Figure 9.

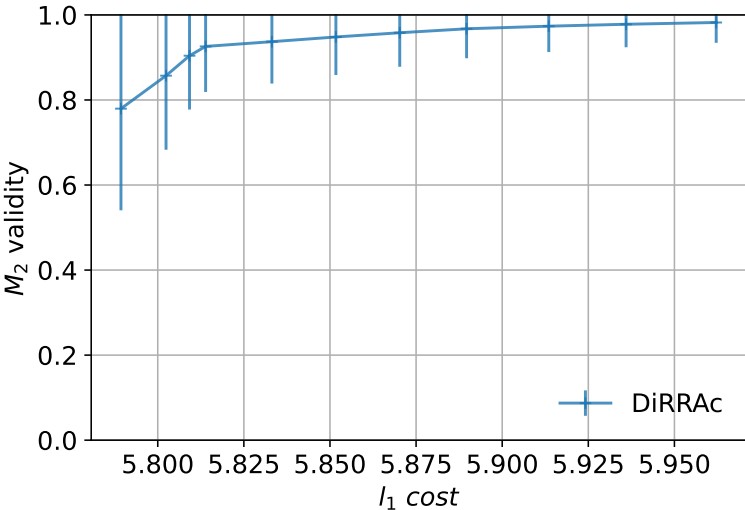

Figure 8: Cost of robustness of DiRRAc

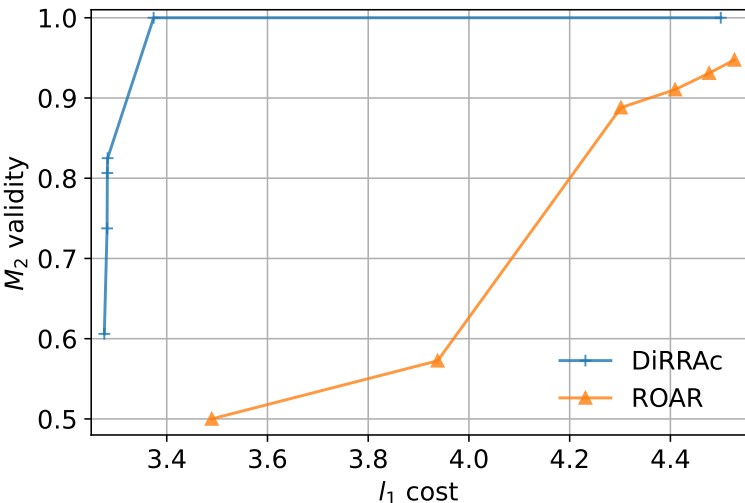

Figure 9: Comparison of $M_2$ validity as a function of the $l_1$ distance between input instance and the recourse for our DiRRAc method and ROAR

**Real-world data.** For each of three real-world datasets, we choose a number of features to benchmark with other baselines:

- For German credit dataset from UCI repository, we choose 5 features: Status, Duration, Credit amount, Personal status, Age. We found in the description of two datasets that feature Status in the data correction shift dataset corrects the coding errors in the original dataset (Dua & Graff, 2017).

- For SBA dataset, following the work from Li et al. (2018) and Upadhyay et al. (2021) we choose the following features: Selected, Term, NoEmp, CreateJob, RetainedJob, UrbanRural, ChgOffPrinGr, GrAppv, SBA_Appv, New, RealEstate, Portion, Recession. We use the instances during 1989-2006 as the original data, and the remaining instances as temporal shift data.

- For Student Performance dataset, motivated by Cortez & Silva (2008), we choose feature G3 - final grade for deciding the label pass or fail for each student. The student who has G3 $<$ 12 is labeled 0 (failed) and 1 (passed) otherwise. For input features, we choose 9 features: Age, Study time, Famsup, Higher, Internet, Health, Absences, G1, G2. We separate the dataset into the original and the geospatial shift data by 2 different schools.

Table 5: Accuracy of the underlying classifiers.

| Dataset | Methods | Accuracy |
|---|---|---|
| German Credit | LR | $0.72 \pm 0.00$ |
| | MLPs | $0.76 \pm 0.01$ |
| Shifted German Credit | LR | $0.7 \pm 0.00$ |
| | MLPs | $0.72 \pm 0.01$ |
| SBA | LR | $0.79 \pm 0.01$ |
| | MLPs | $0.93 \pm 0.02$ |
| Shifted SBA | LR | $0.77 \pm 0.01$ |
| | MLPs | $0.89 \pm 0.01$ |
| Student Performance | LR | $0.84 \pm 0.01$ |
| | MLPs | $0.91 \pm 0.01$ |
| Shifted Student Performance | LR | $0.91 \pm 0.00$ |
| | MLPs | $0.99 \pm 0.01$ |

**Experiments with Euclidean norm.** We accumulate here the numerical results with $c(x, x_0) = \|x - x_0\|_2$. Table 6 reports the $M_2$ validity, along with the $l_1$ and $l_2$ distance between the input instance and its corresponding recourse. Results are mean $\pm$ standard deviation, calculated from 100 independent replications. The settings of this experiments are the same as the experiments in Section 5

Table 6: Benchmark of validity, $l_1$ and $l_2$ using Euclidean cost on different real-world datasets.

| Dataset | Methods | $M_2$ validity | $l_1$ cost | $l_2$ cost |
|---|---|---|---|---|
| German Credit | AR | $0.78 \pm 0.00$ | $\mathbf{1.26} \pm 0.68$ | $0.94 \pm 0.41$ |
| | MACE | $0.97 \pm 0.00$ | $2.10 \pm 0.86$ | $1.20 \pm 0.47$ |
| | DiRRAc | $0.99 \pm 0.02$ | $1.72 \pm 0.49$ | $0.77 \pm 0.19$ |
| | Gaussian DiRRAc | $\mathbf{1.00} \pm 0.00$ | $1.78 \pm 0.49$ | $\mathbf{0.77} \pm 0.19$ |
| SBA | AR | $0.41 \pm 0.13$ | $\mathbf{1.80} \pm 1.14$ | $\mathbf{1.16} \pm 0.60$ |
| | MACE | $0.98 \pm 0.14$ | $3.99 \pm 0.22$ | $1.92 \pm 0.07$ |
| | DiRRAc | $\mathbf{0.98} \pm 0.02$ | $2.43 \pm 1.30$ | $1.17 \pm 0.53$ |
| | Gaussian DiRRAc | $0.92 \pm 0.02$ | $2.43 \pm 1.35$ | $1.18 \pm 0.54$ |
| Student Performance | AR | $0.35 \pm 0.12$ | $1.18 \pm 0.99$ | $0.82 \pm 0.60$ |
| | MACE | $0.64 \pm 0.09$ | $\mathbf{0.81} \pm 0.40$ | $\mathbf{0.51} \pm 0.23$ |
| | DiRRAc | $\mathbf{1.00} \pm 0.00$ | $1.30 \pm 0.38$ | $0.69 \pm 0.16$ |
| | Gaussian DiRRAc | $1.00 \pm 0.00$ | $1.32 \pm 0.40$ | $0.71 \pm 0.16$ |

Table 7: Parameters for the experiments with real-world data in Table 6

| Parameters | Values |
|---|---|
| $K$ | 1 |
| $\delta_{\mathrm{add}}$ | 0.5 |
| $\widehat{p}$ | [1] |
| $\rho$ | [0] |
| $\lambda$ | 0.7 |
| $\zeta$ | 1 |

**Experiments with prior on $\widehat{\Sigma}$.**

In this experiments we assume that we does not have access to the training data. We set the $\widehat{\theta}_1 = \theta_0$, $\theta_0$ is parameters of the original classifier. Then we choose $\widehat{\Sigma}_1 = \tau * I$. We generate recourse for each input instance and compute $M_1$ using the original classifier and $M_2$ validity using the shifted classifiers. In this experiments we choose $\tau = 0.1$.

The results in Table 8 show that our methods remain the same performance while keeping the $l_1$ and $l_2$ cost lower than ROAR in three datasets except the $l_2$ cost in SBA dataset.

Table 8: Benchmark of $M_1$ validity, $M_2$ validity, $l_1$ and $l_2$ using $\widehat{\Sigma}_1 = 0.1 * I$ on different real-world datasets.

| Dataset | Methods | $M_1$ validity | $M_2$ validity | $l_1$ cost | $l_2$ cost |
|---|---|---|---|---|---|
| German Credit | ROAR | $1.00 \pm 0.00$ | $\mathbf{1.00} \pm 0.00$ | $2.60 \pm 0.40$ | $1.08 \pm 0.16$ |
| | DiRRAc | $\mathbf{1.00} \pm 0.00$ | $0.97 \pm 0.05$ | $\mathbf{1.97} \pm 0.34$ | $1.06 \pm 0.13$ |
| | Gaussian DiRRAc | $1.00 \pm 0.00$ | $0.98 \pm 0.04$ | $1.97 \pm 0.34$ | $\mathbf{0.97} \pm 0.14$ |
| SBA | ROAR | $1.00 \pm 1.00$ | $1.00 \pm 0.00$ | $2.25 \pm 0.55$ | $\mathbf{0.98} \pm 0.23$ |
| | DiRRAc | $\mathbf{1.00} \pm 0.00$ | $\mathbf{1.00} \pm 0.00$ | $2.13 \pm 0.41$ | $1.44 \pm 0.26$ |
| | Gaussian DiRRAc | $1.00 \pm 0.00$ | $1.00 \pm 0.00$ | $\mathbf{2.11} \pm 0.45$ | $1.27 \pm 0.41$ |
| Student Performance | ROAR | $1.00 \pm 0.01$ | $0.98 \pm 0.02$ | $1.70 \pm 0.27$ | $0.81 \pm 0.13$ |
| | DiRRAc | $\mathbf{1.00} \pm 0.00$ | $\mathbf{1.00} \pm 0.00$ | $1.73 \pm 0.18$ | $1.43 \pm 0.16$ |
| | Gaussian DiRRAc | $1.00 \pm 0.00$ | $1.00 \pm 0.00$ | $\mathbf{1.21} \pm 0.33$ | $\mathbf{0.64} \pm 0.16$ |

## B  PROOFS

### B.1  PROOFS OF SECTION 3

To prove Proposition 3.4, we are using the notion of Value-at-Risk which is formally defined as follows.

**Definition B.1** (Value-at-Risk). *For any fixed distribution $\mathbb{Q}_k$ of $\tilde{\theta}$, the Value-at-Risk at the risk tolerance level $\beta \in (0, 1)$ of the loss $\tilde{\theta}^\top x$ is defined as*

$$\mathbb{Q}_k\text{-}\mathrm{VaR}_\beta(\tilde{\theta}^\top x) \triangleq \inf\{\tau \in \mathbb{R} : \mathbb{Q}_k(\tilde{\theta}^\top x \le \tau) \ge 1 - \beta\}$$

We are now ready to provide the proof of Proposition 3.4.

*Proof of Proposition 3.4.* Using the definition of the Value-at-Risk in Definition B.1, we have

$$\sup_{\mathbb{Q}_k \in \mathcal{B}_k(\widehat{\mathbb{P}}_k)} \mathbb{Q}_k(\tilde{\theta}^\top x \le 0) = \inf\left\{ \beta : \beta \in [0, 1], \sup_{\mathbb{Q}_k \in \mathcal{B}_k(\widehat{\mathbb{P}}_k)} \mathbb{Q}_k(\tilde{\theta}^\top x \le 0) \le \beta \right\}$$

$$= \inf\left\{ \beta : \beta \in [0, 1], \sup_{\mathbb{Q}_k \in \mathcal{B}_k(\widehat{\mathbb{P}}_k)} \mathbb{Q}_k\text{-}\mathrm{VaR}_\beta(-\tilde{\theta}^\top x) \le 0 \right\}$$

By Nguyen (2019, Lemma 3.31), we can reformulate the worst-case value-at-risk as

$$\sup_{\mathbb{Q}_k \in \mathcal{B}_k(\widehat{\mathbb{P}}_k)} \mathbb{Q}_k\text{-}\mathrm{VaR}_\beta(-\tilde{\theta}^\top x) = -\widehat{\theta}_k^\top x + \sqrt{\frac{1-\beta}{\beta}} \sqrt{x^\top \widehat{\Sigma}_k x} + \frac{\rho_k}{\sqrt{\beta}} \|x\|_2.$$

It is now easy to observe that in the first case when $-\widehat{\theta}_k^\top x + \rho_k \|x\|_2 \geq 0$, then we should have $\sup_{\mathbb{Q}_k \in \mathcal{B}_k(\widehat{\mathbb{P}}_k)} \mathbb{Q}_k(\tilde{\theta}^\top x \leq 0) = 1$.

We now consider the second case when $-\widehat{\theta}_k^\top x + \frac{\rho_k}{\sqrt{\beta}} \|x\|_2 < 0$. It is easy to see, by the monotocity of the worst-case value-at-risk with respect to $\beta$, that the minimal value $\beta^\star$ should satisfies

$$-\widehat{\theta}_k^\top x + \sqrt{\frac{1-\beta^\star}{\beta^\star}} \sqrt{x^\top \widehat{\Sigma}_k x} + \frac{\rho_k}{\sqrt{\beta^\star}} \|x\|_2 = 0.$$

Using the transformation $t \leftarrow \sqrt{\beta^\star}$, we have

$$-\widehat{\theta}_k^\top x t + \sqrt{1-t^2} \sqrt{x^\top \widehat{\Sigma}_k x} + \rho_k \|x\|_2 = 0.$$

By rearranging terms and then squaring up both sides, we have the equivalent quadratic equation

$$(A_k^2 + B_k^2) t^2 + 2 A_k C_k t + C_k^2 - B_k^2 = 0$$

with $A_k \triangleq -\widehat{\theta}_k^\top x \leq 0$, $B_k \triangleq \sqrt{x^\top \widehat{\Sigma}_k x} \geq 0$, and $C_k \triangleq \rho_k \|x\|_2 \geq 0$ as defined in the statement of the proposition. Note, moreover, that we also have $A_k^2 \geq C_k^2$. This leads to the solution

$$t = \frac{-A_k C_k + B_k \sqrt{A_k^2 + B_k^2 - C_k^2}}{A_k^2 + B_k^2} \geq 0.$$

Thus, we find

$$f_k(x) = \left( \frac{-A_k C_k + B_k \sqrt{A_k^2 + B_k^2 - C_k^2}}{A_k^2 + B_k^2} \right)^2.$$

This completes the proof. □

We now provide the proof of Theorem 3.2.

*Proof of Theorem 3.2.* We first consider the objective function $f$ of (2), which can be re-expressed as

$$f(x) = \sup_{\mathbb{P} \in \mathcal{B}(\widehat{\mathbb{P}})} \mathbb{P}(\mathcal{C}_{\tilde{\theta}}(x) = 0) = \sup_{\mathbb{Q}_k \in \mathcal{B}_k(\widehat{\mathbb{P}}_k) \, \forall k} \sum_{k \in [K]} \widehat{p}_k \mathbb{Q}_k(\tilde{\theta}^\top x \leq 0)$$

$$= \sum_{k \in [K]} \widehat{p}_k \times \sup_{\mathbb{Q}_k \in \mathcal{B}_k(\widehat{\mathbb{P}}_k)} \mathbb{Q}_k(\tilde{\theta}^\top x \leq 0)$$

$$= \sum_{k \in [K]} \widehat{p}_k \times f_k(x),$$

where the equality in the second line follows from the non-negativity of $\widehat{p}_k$, and the last equality follows from the definition of $f_k(x)$ in (5). Applying Proposition 3.4, we obtain the objective function of problem (4).

Consider now the last constraint of (2). Using the result of Proposition 3.4, this constraint is equivalent to

$$-\widehat{\theta}_k^\top x + \rho_k \|x\|_2 < 0 \qquad \forall k \in [K].$$

This leads to the feasible set $\mathcal{X}$ as is defined in (3). This completes the proof. □

## B.2 Proofs of Section 4

To prove Theorem 4.1, we first define the following worst-case Gaussian component probability function

$$f_k^{\mathcal{N}}(x) \triangleq \sup_{\mathbb{Q}_k \in \mathcal{B}_k^{\mathcal{N}}(\widehat{\mathbb{P}}_k)} \mathbb{Q}_k(\mathcal{C}_{\tilde{\theta}}(x) = 0) = \sup_{\mathbb{Q}_k \in \mathcal{B}_k^{\mathcal{N}}(\widehat{\mathbb{P}}_k)} \mathbb{Q}_k(\tilde{\theta}^\top x \le 0) \qquad \forall k \in [K]. \qquad (11)$$

The next proposition provides the reformulation of $f_k^{\mathcal{N}}$.

**Proposition B.2** (Worst-case probability - Gaussian)**.** *For any* $x \in \mathbb{R}^d$, *any* $k \in [K]$ *and any* $(\widehat{\theta}_k, \widehat{\Sigma}_k, \rho_k) \in \mathbb{R}^d \times \mathbb{S}_+^d \times \mathbb{R}_+$, *define the following constants* $A_k \triangleq -\widehat{\tilde{\theta}}_k^\top x$, $B_k \triangleq \sqrt{x^\top \widehat{\Sigma}_k x}$, *and* $C_k \triangleq \rho_k \|x\|_2$. *The following holds:*

*(i) We have* $f_k^{\mathcal{N}}(x) < \frac{1}{2}$ *if and only if* $A_k + C_k < 0$.

*(ii) If* $x$ *satisfies* $f_k^{\mathcal{N}}(x) < \frac{1}{2}$, *then*

$$f_k^{\mathcal{N}}(x) = 1 - \Phi\Big(\frac{A_k^2 - C_k^2}{-A_k B_k + C_k \sqrt{A_k^2 + B_k^2 - C_k^2}}\Big).$$

*Proof of Proposition B.2.* We first prove Assertion (i). Pick any $\mathbb{Q}_k \in \mathcal{B}_k^{\mathcal{N}}(\widehat{\mathbb{P}}_k)$, then $\mathbb{Q}_k$ is a Gaussian distribution $\mathbb{Q}_k \sim \mathcal{N}(\theta_k, \Sigma_k)$, and thus

$$\mathbb{Q}_k(\tilde{\theta}^\top x \le 0) = \Phi\Big(\frac{-\theta_k^\top x}{\sqrt{x^\top \Sigma x}}\Big).$$

Guaranteeing $f_k^{\mathcal{N}}(x) < \frac{1}{2}$ is equivalent to guaranteeing

$$\sup_{\mathbb{G}((\theta_k, \Sigma_k),(\widehat{\theta}_k, \widehat{\Sigma}_k)) \le \rho_k} -\theta_k^\top x \le 0.$$

Note that we also have

$$\sup_{\mathbb{G}((\theta_k, \Sigma_k),(\widehat{\theta}_k, \widehat{\Sigma}_k)) \le \rho_k} -\theta_k^\top x = \sup_{\theta_k: \|\theta_k - \widehat{\theta}_k\|_2 \le \rho_k} -\theta_k^\top x = -\widehat{\theta}_k^\top x + \rho_k \|x\|_2$$

by the properties of the dual norm. This leads to the equivalent condition that $A_k + C_k < 0$.

We now prove Assertion (ii). Using the definition of the Value-at-Risk in Definition B.1, we have

$$\sup_{\mathbb{Q}_k \in \mathcal{B}_k^{\mathcal{N}}(\widehat{\mathbb{P}}_k)} \mathbb{Q}_k(\tilde{\theta}^\top x \le 0) = \inf\left\{\beta : \beta \in [0, \tfrac{1}{2}), \sup_{\mathbb{Q}_k \in \mathcal{B}_k^{\mathcal{N}}(\widehat{\mathbb{P}}_k)} \mathbb{Q}_k(\tilde{\theta}^\top x \le 0) \le \beta\right\}$$

$$= \inf\left\{\beta : \beta \in [0, \tfrac{1}{2}), \sup_{\mathbb{Q}_k \in \mathcal{B}_k^{\mathcal{N}}(\widehat{\mathbb{P}}_k)} \mathbb{Q}_k\text{-VaR}_\beta(-\tilde{\theta}^\top x) \le 0\right\}$$

Using the result from Nguyen (2019, Lemma 3.31), we have

$$\sup_{\mathbb{Q}_k \in \mathcal{B}_k(\widehat{\mathbb{P}}_k)} \mathbb{Q}_k\text{-VaR}_\beta(-\tilde{\theta}^\top x) = -\widehat{\theta}_k^\top x + t\sqrt{x^\top \widehat{\Sigma}_k x} + \rho\sqrt{1+t^2}\|x\|_2 = A_k + B_k t + C_k\sqrt{1+t},$$

with $t = \Phi^{-1}(1 - \beta)$. Taking the infimum over $\beta$ is then equivalent to finding the root of the equation

$$A_k + t B_k + C_k\sqrt{1+t^2} = 0.$$

Using a transformation $\tau = 1/t$, the above equation becomes

$$A_k \tau + B_k + C_k\sqrt{1+\tau^2} = 0$$

with solution

$$\tau = \frac{-A_k B_k + C_k\sqrt{A_k^2 + B_k^2 - C_k^2}}{A_k^2 - C_k^2} > 0.$$

Notice that $A_k + C_k < 0$, and we also have $A_k^2 > C_k^2$, thus $\tau$ is well-defined. The result now follows by noticing that $f_k^{\mathcal{N}}(x) = 1 - \Phi(t) = 1 - \Phi(1/\tau)$. $\qquad\square$

We are now ready to prove Theorem 4.1.

*Proof of Theorem 4.1.* Problem (8) is equivalent to

$$
\begin{aligned}
\min \quad & \sum_{k \in [K]} \widehat{p}_k \times f_k^{\mathcal{N}}(x) \\
\text{s.t.} \quad & c(x, x_0) \leq \delta \\
& f_k^{\mathcal{N}}(x) < \tfrac{1}{2} \qquad \forall k \in [K].
\end{aligned}
$$

Applying Proposition B.2, we obtain the necessary result. $\qquad\square$

## C  Extensions of the DiRRAc Framework

Throughout this section, we explore two extensions of our DiRRAc framework. In Section C.1, we study an additional layer of robustification with respect to the mixture weights $\widehat{p}$. Next, in Section C.2, we consider an alternative formulation of the objective function to minimize the worst-case component probability.

### C.1  Robustification against Mixture Weight Uncertainty

The DiRRAc problem considered in Section 3 only robustifies the component distributions $\widehat{\mathbb{P}}_k$. We now discuss a plausible approach to robustify against the misspecification of the mixture weights $\widehat{p}$. Because the mixture weights should form a probability vector, it is convenient to model the perturbation in the mixture weights using the $\phi$-divergence.

**Definition C.1** ($\phi$-divergence). *Let $\phi : \mathbb{R} \to \mathbb{R}$ be a convex function on the domain $\mathbb{R}_+$, $\phi(1) = 0$, $0 \times \phi(a/0) = a \times \lim_{t \uparrow \infty} \phi(t)/t$ for $a > 0$, and $0 \times \phi(0/0) = 0$. The $\phi$-divergence $\mathbb{D}_\phi$ between two probability vectors $p$, $\widehat{p} \in \mathbb{R}_+^K$ amounts to $\mathbb{D}_\phi(p \parallel \widehat{p}) \triangleq \sum_{k \in [K]} \widehat{p}_k \times \phi(p_k/\widehat{p}_k)$.*

The family of $\phi$-divergences contains many well-known statistical divergences such as the Kullback-Leibler divergence, the Hellinger distance, etc. Further discussion on this family can be found in Pardo (2018). Distributionally robust optimization models with $\phi$-divergence ambiguity set were originally studied in decision-making problems (Ben-Tal et al., 2013; Bayraksan & Love, 2015) and have recently gained attention thanks to their successes in machine learning tasks (Namkoong & Duchi, 2017; Hashimoto et al., 2018; Duchi et al., 2021).

Let $\varepsilon \geq 0$ be a parameter indicating the uncertainty level of the mixture weights. The uncertainty set for the mixture weights is formally defined as

$$
\Delta \triangleq \left\{ p \in [0,1]^K : \mathbb{1}^\top p = 1,\ \mathbb{D}_\phi(p \parallel \widehat{p}) \leq \varepsilon \right\},
$$

which contains all $K$-dimensional probability vectors which are of $\phi$-divergence at most $\varepsilon$ from the nominal weights $\widehat{p}$. The ambiguity set of the mixture distributions that hedge against the weight misspecification is

$$
\mathcal{U}(\widehat{\mathbb{P}}) \triangleq \left\{ \mathbb{Q} :\ \exists p \in \Delta,\ \exists \mathbb{Q}_k \in \mathcal{B}_k(\widehat{\mathbb{P}}_k)\ \forall k \in [K] \text{ such that } \mathbb{Q} \sim (\mathbb{Q}_k, p_k)\ \right\},
$$

where the component sets $\mathcal{B}_k(\widehat{\mathbb{P}}_k)$ are defined as in Section 3. The DiRRAc problem with respect to the ambiguity set $\mathcal{U}(\widehat{\mathbb{P}})$ becomes

$$
\begin{aligned}
\min \quad & \sup_{\mathbb{P} \in \mathcal{U}(\widehat{\mathbb{P}})} \mathbb{P}(\mathcal{C}_{\tilde{\theta}}(x) = 0) \\
\text{s.t.} \quad & c(x, x_0) \leq \delta \\
& \sup_{\mathbb{Q}_k \in \mathcal{B}_k(\widehat{\mathbb{P}}_k)} \mathbb{Q}_k(\mathcal{C}_{\tilde{\theta}}(x) = 0) < 1 \qquad \forall k \in [K].
\end{aligned}
\tag{12}
$$

It is important to note at this point that the feasible set of (12) coincides with the feasible set of (2). Thus, to resolve problem (12), it suffices to analyze the objective function of (12). Given the function $\phi$, we define its conjugate function $\phi^* : \mathbb{R} \to \mathbb{R} \cup \{\infty\}$ by

$$
\phi^*(s) = \sup_{t \geq 0} \{ ts - \phi(t) \}.
$$

The next theorem asserts that the worst-case probability under $\mathcal{U}(\widehat{\mathbb{P}})$ can be computed by solving a convex program.

**Theorem C.2** (Objective value)**.** *The feasible set of problem (12) coincides with $\mathcal{X}$. Further, for every $x \in \mathcal{X}$, the objective value of (12) equals to the optimal value of a convex optimization problem*

$$\sup_{\mathbb{P} \in \mathcal{U}(\widehat{\mathbb{P}})} \mathbb{P}(\mathcal{C}_{\tilde{\theta}}(x) = 0) = \min_{\lambda \in \mathbb{R}_+, \, \eta \in \mathbb{R}} \eta + \varepsilon\lambda + \lambda \sum_{k \in [K]} \widehat{p}_k \phi^* \Big( \frac{f_k(x) - \eta}{\lambda} \Big),$$

*where $f_k(x)$ are computed using Proposition 3.4.*

*Proof of Theorem C.2.* From the definition of the set $\mathcal{U}(\widehat{\mathbb{P}})$, we can rewrite $F$ using a two-layer decomposition

$$
\begin{aligned}
F(x) = \sup_{\mathbb{P} \in \mathcal{U}(\widehat{\mathbb{P}})} \mathbb{P}(\mathcal{C}_{\tilde{\theta}}(x) = 0) &= \sup_{p \in \Delta} \sup_{\mathbb{Q}_k \in \mathcal{B}_k(\widehat{\mathbb{P}}_k) \, \forall k} \sum_{k \in [K]} p_k \mathbb{Q}_k(\tilde{\theta}^\top x \leq 0) \\
&= \sup_{p \in \Delta} \sum_{k \in [K]} p_k \times \sup_{\mathbb{Q}_k \in \mathcal{B}_k(\widehat{\mathbb{P}}_k)} \mathbb{Q}_k(\tilde{\theta}^\top x \leq 0) \\
&= \sup_{p \in \Delta} \sum_{k \in [K]} p_k \times f_k(x),
\end{aligned}
$$

where the equality in the second line follows from the non-negativity of $p_k$, and the last equality follows from the definition of $f_k(x)$ in (5). By applying the result from Ben-Tal et al. (2013, Corollary 4.2), we have

$$
F(x) = \begin{cases} \min & \eta + \varepsilon\lambda + \lambda \sum_{k \in [K]} \widehat{p}_k \phi^* \Big( \frac{f_k(x) - \eta}{\lambda} \Big) \\ \text{s.t.} & \lambda \in \mathbb{R}_+, \, \eta \in \mathbb{R}. \end{cases}
$$

The proof is complete. $\qquad\square$

From the result of Theorem C.2, we can derive the gradient of the objective function of (12) using Danskin's theorem (Shapiro et al., 2009, Theorem 7.21), or simply using auto-differentiation. Furthermore, $\phi^*$ is convex, and thus solving the minimization problem in Theorem C.2 can be done efficiently using convex optimization algorithms.

## C.2    MINIMIZING THE WORST-CASE COMPONENT PROBABILITY

Instead of minimizing the (total) probability of unfavorable outcome, we can consider an alternative formulation where the recourse action minimizes the worst-case *conditional* probability of unfavorable outcome over all $K$ components. Mathematically, if we opt for the component ambiguity sets $\mathcal{B}_k(\widehat{\mathbb{P}}_k)$ constructed in Section 3, then we can solve

$$
\begin{aligned}
\min \quad & \max_{k \in [K]} \sup_{\mathbb{Q}_k \in \mathcal{B}_k(\widehat{\mathbb{P}}_k)} \mathbb{Q}_k(\mathcal{C}_{\tilde{\theta}}(x) = 0) \\
\text{s.t.} \quad & c(x, x_0) \leq \delta \\
& \sup_{\mathbb{Q}_k \in \mathcal{B}_k(\widehat{\mathbb{P}}_k)} \mathbb{Q}_k(\mathcal{C}_{\tilde{\theta}}(x) = 0) < 1 \qquad \forall k \in [K].
\end{aligned}
\tag{13a}
$$

Interestingly, problem (13a) does not involve the mixture weighs $\widehat{p}$. As a consequence, a trivial advantage of this model is that it hedges automatically against the misspecification of $\widehat{p}$. To complete, we provide its equivalent finite-dimensional form.

**Corollary C.3** (Component Probability DiRRAc)**.** *Problem (13a) is equivalent to*

$$
\min_{x \in \mathcal{X}} \max_{k \in [K]} \frac{\rho_k \widehat{\theta}_k^\top x \|x\|_2 + \sqrt{x^\top \widehat{\Sigma}_k x} \sqrt{(\widehat{\theta}_k^\top x)^2 + x^\top \widehat{\Sigma}_k x - \rho_k^2 \|x\|_2^2}}{(\widehat{\theta}_k^\top x)^2 + x^\top \widehat{\Sigma}_k x}.
\tag{13b}
$$

## D    EXTENSIONS OF THE GAUSSIAN DiRRAc FRAMEWORK

In this section, we leverage the results in Section C to extend the Gaussian DiRRAc framework to (i) handle the uncertainty of the mixture weight and (ii) minimize the worst-case modal probability. Remind that each individual mixture ambiguity set $\mathcal{B}_k^{\mathcal{N}}(\widehat{\mathbb{P}}_k)$ is of the form

$$\mathcal{B}_k^{\mathcal{N}}(\widehat{\mathbb{P}}_k) = \left\{ \mathbb{Q}_k : \mathbb{Q}_k \sim \mathcal{N}(\theta_k, \Sigma_k),\ \mathbb{G}((\theta_k, \Sigma_k), (\widehat{\theta}_k, \widehat{\Sigma}_k)) \le \rho_k \right\},$$

which is a ball in the space of Gaussian distributions.

### D.1    HANDLING MIXTURE WEIGHT UNCERTAINTY - GAUSSIAN DiRRAc

Following the notations in Section C.1, we define the set of possible mixture weights as

$$\Delta = \left\{ p \in [0,1]^K : \mathbb{1}^\top p = 1,\ \mathbb{D}_\phi(p \| \widehat{p}) \le \varepsilon \right\}$$

and the ambiguity set with Gaussian information is defined as

$$\mathcal{U}^{\mathcal{N}}(\widehat{\mathbb{P}}) = \left\{ \mathbb{Q} \ :\ \exists p \in \Delta,\ \exists \mathbb{Q}_k \in \mathcal{B}_k^{\mathcal{N}}(\widehat{\mathbb{P}}_k)\ \forall k \in [K]\ \text{such that}\ \mathbb{Q} \sim (\mathbb{Q}_k, p_k)_{k \in [K]} \right\}.$$

The distributionally robust problem with respect to the ambiguity set $\mathcal{U}(\widehat{\mathbb{P}})$ is

$$\begin{aligned}
\inf\ &\sup_{\mathbb{P} \in \mathcal{U}^{\mathcal{N}}(\widehat{\mathbb{P}})} \mathbb{P}(\mathcal{C}_{\tilde{\theta}}(x) = 0) \\
\text{s.t.}\ &c(x, x_0) \le \delta \\
&\sup_{\mathbb{Q}_k \in \mathcal{B}_k^{\mathcal{N}}(\widehat{\mathbb{P}}_k)} \mathbb{Q}_k(\mathcal{C}_{\tilde{\theta}}(x) = 0) < \tfrac{1}{2} \qquad \forall k \in [K].
\end{aligned} \tag{14}$$

Following the results in Section 4, the feasible set of (14) coincides with the set $\mathcal{X}$. It suffices now to provide the reformulation for the objective function of (14).

**Corollary D.1.** *For any $x \in \mathcal{X}$, we have*

$$\sup_{\mathbb{P} \in \mathcal{U}^{\mathcal{N}}(\widehat{\mathbb{P}})} \mathbb{P}(\mathcal{C}_{\tilde{\theta}}(x) = 0) = \begin{cases} \inf & \eta + \varepsilon\lambda + \lambda \sum_{k \in [K]} \widehat{p}_k \phi^*\left( \dfrac{f_k^{\mathcal{N}}(x) - \eta}{\lambda} \right) \\ \text{s.t.} & \lambda \in \mathbb{R}_+,\ \eta \in \mathbb{R}, \end{cases}$$

*where the values $f_k^{\mathcal{N}}(x)$ are obtained in Proposition B.2.*

Corollary D.2 follows from Theorem D.2 by replacing the quantities $f_k(x)$ by $f_k^{\mathcal{N}}(x)$ to take into account the Gaussian parametric information. The proof of Corollary D.2 is omitted.

### D.2    MINIMIZING WORST-CASE COMPONENT PROBABILITY

We now consider the Gaussian DiRRAc that minimizes the worst-case modal probability of infeasibility. More concretely, we consider the recourse action obtained by solving

$$\begin{aligned}
\inf\ &\max_{k \in [K]}\ \sup_{\mathbb{Q}_k \in \mathcal{B}_k^{\mathcal{N}}(\widehat{\mathbb{P}}_k)} \mathbb{Q}_k(\mathcal{C}_{\tilde{\theta}}(x) = 0) \\
\text{s.t.}\ &c(x, x_0) \le \delta \\
&\sup_{\mathbb{Q}_k \in \mathcal{B}_k^{\mathcal{N}}(\widehat{\mathbb{P}}_k)} \mathbb{Q}_k(\mathcal{C}_{\tilde{\theta}}(x) = 0) < \tfrac{1}{2} \qquad \forall k \in [K].
\end{aligned} \tag{15a}$$

The next corollary provides the equivalent form of the above optimization problem.

**Corollary D.2.** *Problem (15a) is equivalent to*

$$\inf_{x \in \mathcal{X}}\ \max_{k \in [K]} \left\{ 1 - \Phi\left( \frac{(\widehat{\theta}_k^\top x)^2 - \rho_k^2 \|x\|_2^2}{\widehat{\theta}_k^\top x \sqrt{x^\top \widehat{\Sigma}_k x} + \rho_k \|x\|_2 \sqrt{(\widehat{\theta}_k^\top x)^2 + x^\top \widehat{\Sigma}_k x - \rho_k^2 \|x\|_2^2}} \right) \right\}. \tag{15b}$$

---

**Algorithm 1** Projected gradient descent algorithm with backtracking line-search

---

**Input:** Input instance $x_0$, feasible set $\mathcal{X}_\varepsilon$ and objective function $f$
**Line search parameters:** $\lambda \in (0,1)$, $\zeta > 0$ (Default values: $\lambda = 0.7, \zeta = 1$)
**Initialization:** Set $x^0 \leftarrow \mathrm{Proj}_{\mathcal{X}_\varepsilon}(x_0)$
**for** $t = 0, \ldots, T-1$ **do**
   Find the smallest integer $i \geq 0$ such that

$$f\left(\mathrm{Proj}_{\mathcal{X}_\varepsilon}(x^t - \lambda^i \zeta \nabla f(x^t))\right) \leq f(x^t) - \frac{1}{2\lambda^i \zeta} \|x^t - \mathrm{Proj}_{\mathcal{X}_\varepsilon}(x^t - \lambda^i \zeta \nabla f(x^t))\|_2^2.$$

   Set $x^{t+1} = \mathrm{Proj}_{\mathcal{X}_\varepsilon}(x^t - \lambda^i \zeta \nabla f(x^t))$.
**end for**
**Output:** $x^T$

---

# E    PROJECTED GRADIENT DESCENT ALGORITHM

The pseudocode of the algorithm is presented in Algorithm 1. The convergence guarantee for Algorithm 1 follows from Beck (2017, Theorem 10.15), and is distilled in the next theorem.

**Theorem E.1** (Convergence guarantee)**.** *Let $\{x^t\}_{t=0,1,\ldots,T}$ be the sequence generated by Algorithm 1. Then, all limit points of the sequence $\{x^t\}_{t=0,1,\ldots,T}$ are stationary points of problem (4) with the modified feasible set $\mathcal{X}_\varepsilon$. Furthermore, there exists some constant $C > 0$ such that for any $T \geq 1$, we have*

$$\min_{t=0,1,\ldots,T} \frac{\left\|x^t - \mathrm{Proj}_{\mathcal{X}_\varepsilon}\left(x^t - \zeta \nabla f(x^t)\right)\right\|_2}{\zeta} \leq \frac{C}{\sqrt{T}}.$$

