# OpenReview forum: "Distributionally Robust Recourse Action"
_ICLR.cc/2022/Conference — ICLR 2022 Submitted_

### Official Review · Reviewer_a91J · 2021-10-29

**Correctness:** 3
**Technical Novelty And Significance:** 3
**Empirical Novelty And Significance:** 1
**Recommendation:** 3
**Confidence:** 5

**Main Review:**

Strengths: very clear and thorough exposition of the approach proposed. The approach proposed is very sound technically.

Weaknesses: the experiments presented are rather limited and the value of the contributions is unclear. In particular, the problem of generating robust recourse was previously considered by Upadhyay et al., 2021, but the authors do not provide any evidence as to why their approach may be preferable, which is particularly concerning given that the experiments considered by the authors are heavily inspired in those of Upadhyay et al., 2021.

To strengthen the contributions of the paper, the authors should focus on improving the experiments section. The contribution of this paper would be much stronger if the authors compared the performance of their approach to that of Upadhyay et al., 2021, validated the claim that “robust optimization solutions can be overly conservative because it may hedge against a pathological parameter in the uncertainty set” in the context of algorithmic recourse, and showed that their proposed approach overcomes this issue.

Detailed comments on the experiments section:
* As previously mentioned, authors should compare their approach to Upadhyay et al., 2021.
* Why was L1 not used as the cost function similarly to AR and MACE? Is this an inherent limitation of DiRRAc? If possible, it would be best to use L1 as the cost function for all three approaches.
* It would be very valuable to also include experiments for non-linear classifiers (e.g. MLPs).
* It would be very valuable to consider actionability constraints. It seems like in practice many of the features of the real-world data sets would be immutable (e.g. “Recession” in the SBA data set).
* For the real-world data, you use significantly fewer features than Upadhyay et al., 2021, why is this?
* In my opinion, none of the test data for which recourse is generated should be used for estimating theta and Sigma. My suggested approach: split the data into train and test, train 100 classifiers only with the train data (e.g. by subsampling 80% of the train data) to estimate theta and Sigma, and then compute recourse on the test data.
* In the recourse setting, one typically does not assume access to the training data. It would be valuable to discuss how could theta and Sigma be estimated (or rather guessed) given no access to the data, and what would be the implications in terms of recourse validity. This could also be tested experimentally by setting theta to be the weights of the classifier for which recourse is generated and over and under estimating Sigma to various degrees.
* For the real-world data, please include the accuracy of the classifier for which recourse is generated.


Detailed comments on the introduction:
* Contrary to what is stated in  Section 1 Paragraph 2, in my opinion “counterfactual explanations” and “recourse actions” should not be used interchangeably, but rather authors should refer only to “recourse actions”, since the problem of recourse validity under data shifts, at least how it is presented in this work (Section 1 Paragraph 5), is not applicable to counterfactual explanations.
* Section 1, Paragraph 2: “If a specific application can provide the negative outcomes with recourse actions, it can improve the user engagement and boost the interpretability at the same time.” Citations needed for these statements.
* Section 1, Paragraph 3: “we must consider age as an immutable feature” It is not a must to consider age immutable, several works consider it as a non-decreasing feature.
* Section 1, Paragraph 4: “Various solutions has have been proposed...”
* Section 1, Paragraph 5, “Data shifts usually induce corresponding shifts in the machine learning models’ parameters” Organizations usually retrain models periodically in repose to data shifts. The data shift itself does not induces changes to the parameters of the model.
* Section 1, Paragraph 5: “If a recourse action fails to generate a favorable outcome in the future, then the recourse action becomes useless” Not necessarily, see Handling change over time via ex post facto in Venkatasubramanian and Alfano, 2020.
* Section 1, Paragraph 5: “and the trust on the machine learning system is lost.” Citation needed.

Comments regarding section 2, 3 and 4:
* In my opinion Algorithm 1 and Theorem 3.4 should not be in the main text, as these are just general well-known concepts from convex optimization, leaving more room for the experiments sections. Authors could just mention convergence rate of $O(1/\sqrt(T))$.
* Similarly, Sections 4.1 and 4.2 could be moved to the appendix, since they are not core to the main contribution of the paper and are not evaluated in the experiments section. This would also give the authors more space for the experiments section and a conclusion.
* Equation 1: min_x
* Equation 2, 7, 8a, 9a: $inf/min_{x \in X}$ or add $x \in X$ as an explicit constrain similarly to Equation 1.
* Equation 9a, please explain why a margin of 0.5 rather than 1 is used.


**Summary Of The Paper:**

The paper considers the problem of generating recourse actions that are robust to shifts in the parameters of the classifier. The authors present a distributionally robust optimization approach and experimentally show that for linear classifiers and under no actionability constraints, the approach generates recourse actions that have high probability of being valid under shifts to the weights of the linear classifier.

**Summary Of The Review:**

While the authors present a very clear and thorough exposition of the approach proposed and the approach is technically sound, the experiments presented are very limited and the overall value of the contribution is unclear. In particular, the authors do not compare their proposed method with previous approaches addressing the problem of generating robust recourse actions.

---

> ### Author Response · Authors · 2021-11-10
> **Comparison with ROAR**
>
> Dear Reviewer,
>
> We would like to clarify that the ROAR method proposed by Upadhyay et al. (2021) is accepted at NeurIPS 2021. The ICLR 2022 guidelines indicate that our paper and ROAR are contemporaneous, and no comparison is required. (Edited on Nov 10: As of this date, the code for ROAR is still not publicly available -- this also hinders our comparison)
>
> For further details, the guideline can be found at the following link: https://iclr.cc/Conferences/2022/ReviewerGuide . The exact Q&A is quoted below.
>
> Q: Are authors expected to cite and compare with very recent work? What about non peer-reviewed (e.g., ArXiv) papers?
> A: We consider papers contemporaneous if they are published (available in online proceedings) within the last four months. That means, since our full paper deadline is October 5, if a paper was published (i.e., at a peer-reviewed venue) on or after June 5, 2021, authors are not required to compare their own work to that paper. Authors are encouraged to cite and discuss all relevant papers, but they may be excused for not knowing about papers not published in peer-reviewed conference proceedings or journals, which includes papers exclusively available on arXiv. Reviewers are encouraged to use their own good judgement and, if in doubt, discuss with their area chair.
>
>
> The relevant paper is submitted to arXiv in February 2021, and is later accepted to NeurIPS 2021. The full reference is:
>
> Sohini Upadhyay, Shalmali Joshi, and Himabindu Lakkaraju. Towards robust and reliable algorithmic recourse. arXiv preprint arXiv:2102.13620, 2021.
>
> We truly appreciate if you can take this information into account. Thank you very much for your consideration!

---

> ### Comment · Area_Chair_pWJ1 · 2021-11-18
> **Response to authors' comment**
>
> Hi,
>
> could you respond to the authors' comment and say if it changes your review?

---

> ### Author Response · Authors · 2021-11-22
> **Response to Review [Part 1]**
>
> We thank the reviewer for the detailed review and suggestions. Our responses to the questions are below.
>
> **Experiments section:**
> * **Comparison with ROAR.** We have implemented the RObust Algorithmic Recourse (ROAR) framework. We have a comparison to ROAR with linear model on the real-world datasets. The following table demonstrated that our framework has a higher validity, while keeping the low cost compared to ROAR in all three datasets. We have included the results in our revised draft (Table 1). In the experiment with synthetic data, we evaluate the trade-offs between $l_{1}$ cost and validity of recourses generated by DiRRAc and ROAR. The experiment shows that even at a small cost, recourses generated by our framework are robust to model shifts while ROAR requires much higher cost to obtain a high validity. The results of this experiment are provided in our revised draft (Figure 9).
>
> * **Experiments with L1 cost.**  $l_{1}$ cost is not a limitation of our framework. The table below show that our framework using $l_{1}$ as the cost function achieves high validity and low cost compared to other baselines.
>
> | Dataset  |Methods               |$M_{1}$ validity                |$M_{2}$ validity               |$l_{1}$ cost               |$l_{2}$ cost     |
> |----------|-----------------------------------------|---------------------|---------------------|---------------------|---------------------|
> | German Credit| AR | 0.73 $\pm$ 0.25    | 0.78 $\pm$ 0.00    | 1.26 $\pm$ 0.68    |0.94 $\pm$ 0.41    |
> |          |MACE    |0.87 $\pm$ 0.15|0.97 $\pm$ 0.00    |2.11 $\pm$ 0.86|1.20 $\pm$ 0.47 |
> |          |ROAR    |1.00 $\pm$ 0.00    |1.00 $\pm$ 0.00    |2.60 $\pm$ 0.40    |1.08 $\pm$ 0.16   |
> |          |DiRRAc  |**1.00** $\pm$ 0.00    |**1.00** $\pm$ 0.00|2.09 $\pm$ 0.43    |0.96 $\pm$ 0.18    |
> |          |Gaussian DiRRAc |1.00 $\pm$ 0.00    |0.93 $\pm$ 0.05|**0.73** $\pm$ 0.47    |**0.47** $\pm$ 0.38    |
> | SBA| AR   | 0.26 $\pm$ 0.24 | 0.42 $\pm$ 0.14 | 3.41 $\pm$ 2.10 | 1.56 $\pm$ 0.76 |
> |          |MACE    |1.00 $\pm$ 0.00 |  1.00 $\pm$ 0.00 | 6.85 $\pm$ 0.56 | 2.50 $\pm$ 0.11 |
> |          |ROAR    |1.00 $\pm$ 1.00 |  1.00 $\pm$ 0.00 |   2.25 $\pm$ 0.55 |   0.98 $\pm$ 0.23 |
> |          |DiRRAc  |**1.00** $\pm$ 0.00 |  **1.00** $\pm$ 0.00|    **1.13** $\pm$ 0.43|    **0.82** $\pm$ 0.31 |
> |          |Gaussian DiRRAc |1.00 $\pm$ 0.01|   1.00 $\pm$ 0.00|    1.14 $\pm$ 0.42|    0.83 $\pm$ 0.30|
> | Student Performance| AR   | 0.28 $\pm$ 0.08|  0.35 $\pm$ 0.12|    1.18 $\pm$ 0.99|    0.82 $\pm$ 0.60    |
> |          |MACE    |0.66 $\pm$ 0.12    |0.57 $\pm$ 0.10    |0.81 $\pm$ 0.40|   **0.51** $\pm$ 0.24|
> |          |ROAR    |1.00 $\pm$ 0.01|0.98 $\pm$ 0.02|   1.70 $\pm$ 0.27|    0.81 $\pm$ 0.13|
> |          |DiRRAc  |**1.00** $\pm$ 0.00 | **0.99** $\pm$ 0.02| **0.74** $\pm$ 0.18 |0.63 $\pm$ 0.14 |
> |          |Gaussian DiRRAc |1.00 $\pm$ 0.00|0.98 $\pm$ 0.02|   0.74 $\pm$ 0.18|    0.74 $\pm$ 0.18|
>
> * **Actionability constraints**. Our framework supports actionability constraints by adding more constraints to the projection by hand but in this work, we are not considering those constraints for real data since we are not experts in finance or education. Besides, some features are very hard to specify as immutable features or actionable features and we need to define those constraints case by case. In real-world applications, we suppose that users should define those constraints.
>
> * **Fewer features than ROAR**. In the experiment with the German Credit dataset, we use more features than ROAR (5 features compared to 4 features in ROAR). In other datasets, we are motivated by 2 papers (Li et al. [4], Cortez & Silva [5]) that do data mining tasks on SBA and Student Performance dataset. We choose the features that are claimed to be explanatory variables in those papers. For the SBA dataset, we have realized that using more features can significantly improve the accuracy of the classifiers. So in the revised version, we use more features: ‘Selected’, ‘Term’, ‘NoEmp’, ‘CreateJob’, ‘RetainedJob’, ‘UrbanRural’, ‘ChgOffPrinGr’, ‘GrAppv’, ‘SBA_Appv’, ‘New’, ‘RealEstate’, ‘Portion’, ‘Recession’.
>
> * **Estimating theta and sigma.** In our experiments, we only use training data for estimating theta and sigma. Our approach for estimating theta and sigma is similar to your suggestions: we split randomly 80% of the original dataset and train a logistic classifier on the training data. This process is repeated independently 100 times to obtain 100 observations of the model parameters, then we compute the empirical mean and covariance matrix for $\hat{\theta_{1}}$ and  $\hat{\Sigma_{1}}$. It is guaranteed that the test data is not used for estimating the nominal moments.

---

> > ### Author Response · Authors · 2021-11-23
> > **Response to Review [Part 2]**
> >
> > **Experiments section:**
> > * **Experiments for non-linear classifiers.** Following the previous work ( Upadhyay et al. [1], Rawal et al. [2]) , we have extended our method to non-linear classifiers by using LIME (Ribeiro et al. [3]) to generate the local approximation linear model for MLPs classifier. The results show that our framework can increase empirical validity significantly while keeping a low cost. We have included the results in our revised draft (Table 2). We report the results below.
> > | Dataset  |Methods               | $M_{1}$ validity                | $M_{2}$ validity               | $l_{1}$ cost                 | $l_{2}$ cost     |
> > |----------|-----------------------------------------|---------------------|---------------------|---------------------|---------------------|
> > | German Credit| AR | 0.67 $\pm$ 0.47    | 0.59 $\pm$ 0.38    | **1.00** $\pm$ 0.00    |1.00 $\pm$ 0.00    |
> > |          |MACE    |0.67 $\pm$ 0.47|0.31 $\pm$ 0.22    |1.99 $\pm$ 0.29|1.19 $\pm$ 0.13 |
> > |          |ROAR    |0.87 $\pm$ 0.26    |0.66 $\pm$ 0.33    |2.66 $\pm$ 0.16    |1.21 $\pm$ 0.01   |
> > |          |DiRRAc  |**1.00** $\pm$ 0.00    |**0.80** $\pm$ 0.18|1.07 $\pm$ 0.01    |1.00 $\pm$ 0.01    |
> > |          |Gaussian DiRRAc |0.91 $\pm$ 0.29    |0.73 $\pm$ 0.29|1.05 $\pm$ 0.07    |**0.95** $\pm$ 0.17    |
> > | SBA| AR   | 1.00 $\pm$ 0.00 | 0.72 $\pm$ 0.27 | **1.02** $\pm$ 0.04 | 1.00 $\pm$ 0.00 |
> > |          |MACE    |1.00 $\pm$ 0.00 |  0.75 $\pm$ 0.16 | 5.90 $\pm$ 0.45 | 2.31 $\pm$ 0.07 |
> > |          |ROAR    |1.00 $\pm$ 0.00 |  0.91 $\pm$ 0.29 |   3.34 $\pm$ 0.22 |   1.07 $\pm$ 0.07 |
> > |          |DiRRAc  |**1.00** $\pm$ 0.00 |  **0.97** $\pm$ 0.09|    1.07 $\pm$ 0.03|    **0.73** $\pm$ 0.11 |
> > |          |Gaussian DiRRAc |1.00 $\pm$ 0.00 |  0.85 $\pm$ 0.23 |1.07 $\pm$ 0.03|   0.81 $\pm$ 0.09|
> > | Student Performance| AR   | 0.47 $\pm$ 0.50   |0.41 $\pm$ 0.44|   1.02 $\pm$ 0.02|    1.02 $\pm$ 0.02|
> > |          |MACE    |0.60 $\pm$ 0.49|   0.60 $\pm$ 0.49 |3.04 $\pm$ 1.61|   1.44 $\pm$ 0.57|
> > |          |ROAR    |0.95 $\pm$ 0.22|   0.86 $\pm$ 0.29|    3.67 $\pm$ 0.60|    1.24 $\pm$ 0.17|
> > |          |DiRRAc  |**1.00** $\pm$ 0.00 | 0.94 $\pm$ 0.17| **0.95** $\pm$ 0.56 |**0.84** $\pm$ 0.26 |
> > |          |Gaussian DiRRAc |1.00 $\pm$ 0.00|**0.96** $\pm$ 0.16|   0.95 $\pm$ 0.56|    0.87 $\pm$ 0.30|
> > * **Without access to the training data.** The reviewer is touching on a relevant point, and we also agree that in practice, the training data is not available. In this case, we need to put stronger prior on the nominal moments $\hat \theta$ and $\hat \Sigma$. With minimal assumption, we can choose $\hat{\theta}_1$ to be the weights of the classifier for which recourse is generated and set $\hat{\Sigma}_1$ as an isotropic covariance matrix.  We demonstrate that this choice of nominal moments works well in practice. Towards this end, we choose $\hat{\Sigma}_1 = 0.1 I$ and rerun our (Gaussian) DiRRAc model on real datasets. We add these results in Appendix A of our revised manuscript.
> > | Dataset  |Methods               | $M_{1}$ validity                | $M_{2}$ validity               | $l_{1}$ cost                 | $l_{2}$ cost     |
> > |----------|-----------------------------------------|---------------------|---------------------|---------------------|---------------------|
> > |  German Credit        |ROAR    |1.00 $\pm$ 0.00    |**1.00** $\pm$ 0.00    |2.60 $\pm$ 0.40    |1.08 $\pm$ 0.16   |
> > |          |DiRRAc    |**1.00** $\pm$ 0.00    |0.97 $\pm$ 0.05|**1.97** $\pm$ 0.34    |1.06 $\pm$ 0.13    |
> > |          |Gaussian DiRRAc |1.00 $\pm$ 0.00    |0.98 $\pm$ 0.04|1.97 $\pm$ 0.34    |**0.97** $\pm$ 0.14    |
> > | SBA         |ROAR    |1.00 $\pm$ 1.00 |  1.00 $\pm$ 0.00 |   2.25 $\pm$ 0.55 |   **0.98** $\pm$ 0.23 |
> > |          |DiRRAc  |**1.00** $\pm$ 0.00 |  **1.00** $\pm$ 0.00| 2.13 $\pm$ 0.41 | 1.44 $\pm$ 0.26 |
> > |          |Gaussian DiRRAc |1.00 $\pm$ 0.00 |  1.00 $\pm$ 0.00 |**2.11** $\pm$ 0.45|   1.27 $\pm$ 41|
> > | Student Performance         |ROAR    |1.00 $\pm$ 0.01|0.98 $\pm$ 0.02|   1.70 $\pm$ 0.27|    0.81 $\pm$ 0.13|
> > |          |DiRRAc  |**1.00** $\pm$ 0.00 | **1.00** $\pm$ 0.00| 1.83 $\pm$ 0.18 |1.43 $\pm$ 0.16 |
> > |          |Gaussian DiRRAc |1.00 $\pm$ 0.00|1.00 $\pm$ 0.00|   **1.21** $\pm$ 0.33|    **0.64** $\pm$ 0.16|

---

> > > ### Author Response · Authors · 2021-11-23
> > > **Response to Review [Part 3]**
> > >
> > > **Experiments section:**
> > > * **Accuracy of the classifier.** The accuracy of linear and non-linear classifiers on original and shifted datasets are included in Table 5 of our revised draft and we report the results below.
> > > | Dataset  |Methods               |Accuracy                |
> > > |----------|-----------------------------------------|---------------------|
> > > | German Credit          |LR    |0.72 $\pm$ 0.00    |
> > > |          |MLPs    |0.76 $\pm$ 0.01    |
> > > | Shifted German Credit          |LR    |0.7 $\pm$ 0.00    |
> > > |          |MLPs    |0.72 $\pm$ 0.01    |
> > > | SBA          |LR  |0.79 $\pm$ 0.01    |
> > > |          |MLPs    |0.93 $\pm$ 0.02    |
> > > | Shifted SBA          |LR  |0.77 $\pm$ 0.01    |
> > > |          |MLPs    |0.89 $\pm$ 0.01    |
> > > | Student Performance          |LR  |0.84 $\pm$ 0.01    |
> > > |          |MLPs    |0.91 $\pm$ 0.01    |
> > > | Shifted Student Performance          |LR  |0.91 $\pm$ 0.00    |
> > > |          |MLPs    |0.99 $\pm$ 0.01    |
> > >
> > > **Introduction section:**
> > > * **Counterfactual explanations and recourse actions.** We agree that in the problem of recourse validity under data shifts, we should only use "recourse actions". We have made the changes in the revised draft.
> > > * **Section 1, Paragraph 2, Citation.** The lack of recourse is often mentioned in calls for increased transparency and explainability in algorithmic decision-making (Ustun et al [7]). Recourse provides explanations and recommendations to individuals who are unfavorably treated by automated decision-making systems (Karimi et al [8]).
> > > * **Section 1, Paragraph 3, Age as an immutable feature.** We agree with the reviewer that age may be considered as a non-decreasing feature in some cases. We have updated this sentence in our revised draft as follows: "we must consider date of birth as an immutable feature".
> > > * **Section 1, Paragraph 4, Minor issue.** Thanks for your comment on this minor issue, we have updated this sentence in our revised version.
> > > * **Section 1, Paragraph 5, The data shift itself does not induce changes to the parameters of the model.** We agree with your comment that data shifts itself does not induce changes to the parameters of the model. We have changed this sentence as follows: "Organizations usually retrain models as a response to data shifts and this induces corresponding shifts in the machine learning models parameters”.
> > > * **Section 1, Paragraph 5, The recourse action becomes useless.** As in Venkatasubramanian & Alfano [6], we agree that generating recourse gains whatever benefit even if the rules for others have changed in the meantime. We have corrected the sentence in our draft.
> > > * **Section 1, Paragraph 5, The trust in the machine learning system is lost.** An inaccurate explanation lowers the trust in the explanation (Rudin [9]).
> > >
> > > **Comments regarding Section 2, 3, 4:**
> > > * **Appendix.** We have moved Algorithm 1, Theorem 3.4, Section 4.1, and 4.2 to the appendix.
> > > * **Equation issues.** We have corrected those equation issues in our revised draft.
> > > * **Equation 9a, please explain why a margin of 0.5 rather than 1 is used.** The detailed reason is revealed in the proof sketch in Section 4.2 in our revised draft.
> > >
> > > [1] Sohini Upadhyay, Shalmali Joshi, and Himabindu Lakkaraju. Towards robust and reliable algorithmic recourse. In Advances in Neural Information Processing Systems 35, 2021.
> > >
> > > [2] Rawal, Kaivalya, Ece Kamar, and Himabindu Lakkaraju. "Can I Still Trust You?: Understanding the Impact of Distribution Shifts on Algorithmic Recourses." arXiv:2012.11788 (2020).
> > >
> > > [3] Ribeiro, Marco Tulio, Sameer Singh, and Carlos Guestrin. "" Why should i trust you?" Explaining the predictions of any classifier." Proceedings of the 22nd ACM SIGKDD international conference on knowledge discovery and data mining. 2016.
> > >
> > > [4] Min Li, Amy Mickel, and Stanley Taylor. “Should this loan be approved or denied?”: A large dataset with class assignment guidelines. Journal of Statistics Education, 26(1):55–66, 2018.
> > >
> > > [5] Paulo Cortez and Alice Silva. Using data mining to predict secondary school student performance. Proceedings of 5th FUture BUsiness TEChnology Conference, 2008.
> > >
> > > [6] Suresh Venkatasubramanian and Mark Alfano. The philosophical basis of algorithmic recourse. In Proceedings of the 2020 Conference on Fairness, Accountability, and Transparency, FAT*’20, pp. 284–293, New York,  NY, USA, 2020. Association for Computing Machinery. ISBN 9781450369367. doi: 10.1145/3351095.3372876. URL https://doi.org/10.1145/3351095.3372876.
> > >
> > > [7] Ustun, Berk, Alexander Spangher, and Yang Liu. "Actionable recourse in linear classification." Proceedings of the Conference on Fairness, Accountability, and Transparency. 2019.
> > >
> > > [8] Amirhossein Karimi, Bernhard Scholkopf,  and Isabel Valera. A survey of algorithmic recourse: Contrastive explanations and consequential recommendations. arXiv preprint arXiv:2010.04050, 2021.
> > >
> > > [9] Cynthia Rudin. Stop explaining black box machine learning models for high stakes decisions and use interpretable models instead, 2019.

---

### Official Review · Reviewer_4adh · 2021-11-03

**Correctness:** 3
**Technical Novelty And Significance:** 3
**Empirical Novelty And Significance:** 2
**Recommendation:** 5
**Confidence:** 3

**Main Review:**

Strengths:
1. Most existing work on recourse actions do not consider model change, so the problem addressed by the paper is relatively new, and it is an important problem since model/data shifts are common in practice.
2. The idea of considering/modelling model shift as a mixture shift of model parameters, and formalizing the problem as a min-max problem.
3. The experiment results demonstrate the superiority of DiRRAc over the methods compared.
4. The paper is well written.

Weaknesses:
1. It is not very clear how challenging it is to adopt the distributionally robust optimization technique for solving the recourse action problem. It would be useful to let readers know clearly that the adoption is non-trivial, which is particularly helpful for readers who are not familiar wi the distributionally robust optimization techinque.
2. From the paper, ROAR is a method for generating counterfactual explanations that are robust to model shifts, but the experiments conducted do not consider ROAR as a baseline.
3. It is not clear how efficient the proposed method is, compared to existing methods.
4. The performance of the proposed method under no model shifts should be evaluated as well.

Other comments:
1. on page 2, there is a typo: Distribut,lly --> Distributionally
2. The paper does not have a Conclusion section.



**Summary Of The Paper:**

This paper studies the problem of recourse actions (a.k.a. counterfactual explanations) while considering data distribution shifts or model shifts. The proposed Distributionally Robust Recourse Action (DiRRAc) framework has the ability to generate valid recourse actions when model parameters shift over time. DiRRAc adopts the distributionally robust optimization technique and the paper proposes a projected gradient descent method to solve the optimization problem. Experiments are conducted with both synthetic and real world data, and the results have shown that DiRRAc methods can generate recourse actions with higher validity than two existing methods.

**Summary Of The Review:**

The paper tackles a very practical and relatively new problem regarding recourse actions. The overall idea seems reasonable to me and the experiments have demonstrated the effectiveness of the proposed method. The paper is also well written.

However, the paper has a few weaknesses as described above. The missing comparison with ROAR is a main concern. The novelty of the proposed method (regarding the adoption of distributionally robust optimization) should be clarified too.

---

> ### Author Response · Authors · 2021-11-10
> **Comparison with ROAR**
>
> Dear Reviewer,
>
> We would like to clarify that the ROAR method proposed by Upadhyay et al. (2021) is accepted at NeurIPS 2021. The ICLR 2022 guidelines indicate that our paper and ROAR are contemporaneous, and no comparison is required. (Edited on Nov 10: As of this date, the code for ROAR is still not publicly available -- this also hinders our comparison)
>
> For further details, the guideline can be found at the following link: https://iclr.cc/Conferences/2022/ReviewerGuide . The exact Q&A is quoted below.
>
> Q: Are authors expected to cite and compare with very recent work? What about non peer-reviewed (e.g., ArXiv) papers?
> A: We consider papers contemporaneous if they are published (available in online proceedings) within the last four months. That means, since our full paper deadline is October 5, if a paper was published (i.e., at a peer-reviewed venue) on or after June 5, 2021, authors are not required to compare their own work to that paper. Authors are encouraged to cite and discuss all relevant papers, but they may be excused for not knowing about papers not published in peer-reviewed conference proceedings or journals, which includes papers exclusively available on arXiv. Reviewers are encouraged to use their own good judgement and, if in doubt, discuss with their area chair.
>
>
> The relevant paper is submitted to arXiv in February 2021, and is later accepted to NeurIPS 2021. The full reference is:
>
> Sohini Upadhyay, Shalmali Joshi, and Himabindu Lakkaraju. Towards robust and reliable algorithmic recourse. arXiv preprint arXiv:2102.13620, 2021.
>
> We truly appreciate if you can take this information into account. Thank you very much for your consideration!

---

> > ### Comment · Reviewer_4adh · 2021-11-22
> > **Thanks**
> >
> > Thanks the authors for the information on comparison with ROAR, which makes good sense to me. However, I still think this is a borderline paper and would to keep my score, given its relatively low empirical novelty.

---

> > > ### Author Response · Authors · 2021-11-22
> > > **Response to Reviewer 4adh**
> > >
> > > Dear Reviewer 4adh,
> > >
> > > We thank you for the constructive review and for the time reviewing our work, and we really hope to have a further discussion with you to see if our latest response solves the concerns.
> > >
> > > We would sincerely appreciate it if you could reply to the most important points in our rebuttal. As suggested by you, we have discussed explicitly the challenge to adopt the distributionally robust optimization technique for solving the recourse action problem. We have added the  results of the comparison with ROAR, the efficient of our framework and the validity of recourse under no model shifts in our rebuttal. We have also included detailed experimental results (Table 1, Figure 9) in our revised draft.
> > >
> > > We genuinely hope you could kindly check our latest response. Thank you!

---

> ### Comment · Area_Chair_pWJ1 · 2021-11-18
> **Respond to authors' comment**
>
> Hi,
>
> could you respond to the authors' comment and say if it changes your review?

---

> ### Author Response · Authors · 2021-11-22
> **Response to Review [Part 1]**
>
> We thank the reviewer for the constructive comments. Our responses to the questions are below.
>
> **Challenging to adopt the distributionally robust optimization technique for solving the recourse action problem.**  Inherently, distributionally robust optimization problems are difficult because the inner supremum problem is infinite-dimensional: it optimizes over the space of probability distributions. More concretely, for our DiRRAc formulation (2), this supremum problem has a feasible set containing all probability measures that satisfy a certain moment constraint prescribed by the Gelbrich distance. This feasible set is rich because it contains continuous (Gaussian, Laplace, etc.), discrete (binominal, etc.) or even mixed (continuous with jumps) distributions. As a consequence, solving the supremum is hard. We now discussed explicitly this difficulty after formulation (2).
>
> Moreover, we also add Remark 6.2 to clarify on the necessity of using the Gelbrich distance. Indeed, one may be tempted to try other metrics to prescribe the set. However, it is not possible to arrive at a closed-form reformulation of the DiRRAc problem as we achieved in Theorem 3.2 and in Theorem 4.1.
>
> **Comparison with ROAR.** We have implemented the RObust Algorithmic Recourse (ROAR) framework. We have a comparison to ROAR with linear model on the real-world datasets. The following table demonstrated that our framework has a higher validity, while keeping the low cost compared to ROAR in all three datasets. We have included the results in our revised draft (Table 1). In the experiments with synthetic data, we evaluate the trade-offs between $l_{1}$ cost and validity of recourses generated by DiRRAc and ROAR. The experiments show that even at a small cost, recourses generated by our framework are robust to model shifts while ROAR requires much higher cost to obtain a high validity. The results of this experiment are provided in our revised draft (Figure 9).
>
> | Dataset  |Methods               |$M_{1}$ validity    |$M_{2}$ validity               |$l_{1}$ cost                 |$l_{2}$ cost     |
> |----------|-----------------------------------------|---------------------|---------------------|---------------------|---------------------|
> | German Credit         |ROAR   |1.00 $\pm$ 0.00    |1.00 $\pm$ 0.00    |2.60 $\pm$ 0.40    |1.08 $\pm$ 0.16   |
> |          |DiRRAc  |**1.00** $\pm$ 0.00    |**1.00** $\pm$ 0.00|2.09 $\pm$ 0.43    |0.96 $\pm$ 0.18    |
> |          |Gaussian DiRRAc |1.00 $\pm$ 0.00    |0.93 $\pm$ 0.05|**0.73** $\pm$ 0.47    |**0.47** $\pm$ 0.38    |
> | SBA          |ROAR    |1.00 $\pm$ 1.00 |  1.00 $\pm$ 0.00 |   2.25 $\pm$ 0.55 |   0.98 $\pm$ 0.23 |
> |          |DiRRAc  |**1.00** $\pm$ 0.00 |  **1.00** $\pm$ 0.00|    **1.13** $\pm$ 0.43|    **0.82** $\pm$ 0.31 |
> |          |Gaussian DiRRAc |1.00 $\pm$ 0.01|   1.00 $\pm$ 0.00|    1.14 $\pm$ 0.42|    0.83 $\pm$ 0.30|
> | Student Performance         |ROAR |1.00 $\pm$ 0.01|0.98 $\pm$ 0.02|   1.70 $\pm$ 0.27|    0.81 $\pm$ 0.13|
> |          |DiRRAc  |**1.00** $\pm$ 0.00 | **0.99** $\pm$ 0.02| **0.74** $\pm$ 0.18 |**0.63** $\pm$ 0.14 |
> |          |Gaussian DiRRAc |1.00 $\pm$ 0.00|0.98 $\pm$ 0.02|   0.74 $\pm$ 0.18|    0.74 $\pm$ 0.18|

---

> > ### Author Response · Authors · 2021-11-23
> > **Response to Review [Part 2]**
> >
> > **The efficient the DiRRAc.** The experiments show that our framework has a low cost compared to ROAR and other baselines.
> >
> > | Dataset  |Methods     |$l_{1}$ cost                 |$l_{2}$ cost     |
> > |----------|-----------------------------------------|---------------------|---------------------|
> > | German Credit| AR |  1.26 $\pm$ 0.68    |0.94 $\pm$ 0.41    |
> > |          |MACE    |2.11 $\pm$ 0.86|1.20 $\pm$ 0.47 |
> > |          |ROAR    |2.60 $\pm$ 0.40    |1.08 $\pm$ 0.16   |
> > |          |DiRRAc  |2.09 $\pm$ 0.43    |0.96 $\pm$ 0.18    |
> > |          |Gaussian DiRRAc |**0.73** $\pm$ 0.47    |**0.47** $\pm$ 0.38    |
> > | SBA| AR   | 3.41 $\pm$ 2.10 | 1.56 $\pm$ 0.76 |
> > |          |MACE    | 6.85 $\pm$ 0.56 | 2.50 $\pm$ 0.11 |
> > |          |ROAR    |2.25 $\pm$ 0.55 |  0.98 $\pm$ 0.23 |
> > |          |DiRRAc  |   **1.13** $\pm$ 0.43|    **0.82** $\pm$ 0.31 |
> > |          |Gaussian DiRRAc |   1.14 $\pm$ 0.42|    0.83 $\pm$ 0.30|
> > | Student Performance| AR   |   1.18 $\pm$ 0.99|    0.82 $\pm$ 0.60    |
> > |          |MACE    |   0.81 $\pm$ 0.40|    **0.51** $\pm$ 0.24|
> > |          |ROAR    |   1.70 $\pm$ 0.27|    0.81 $\pm$ 0.13|
> > |          |DiRRAc  |   **0.74** $\pm$ 0.18 |0.63 $\pm$ 0.14 |
> > |          |Gaussian DiRRAc |   0.74 $\pm$ 0.18|    0.74 $\pm$ 0.18|
> >
> > **The performance of the proposed method under no model shifts.** We agree that the validity of recourses under no model shifts should be evaluated. We have computed the validity of recourse under no model shifts as $M_{1}$ validity in the table below. The results can be found in our updated draft (Table 1).
> >
> > | Dataset  |Methods               |$M_{1}$ validity                |
> > |----------|-----------------------------------------|---------------------|
> > | German Credit| AR | 0.73 $\pm$ 0.25    |
> > |          |MACE    |0.87 $\pm$ 0.15|
> > |          |ROAR    |1.00 $\pm$ 0.00    |
> > |          |DiRRAc  |**1.00** $\pm$ 0.00    |
> > |          |Gaussian DiRRAc |1.00 $\pm$ 0.00    |
> > | SBA| AR   | 0.26 $\pm$ 0.24 |
> > |          |MACE    |1.00 $\pm$ 0.00 |
> > |          |ROAR    |1.00 $\pm$ 1.00 |
> > |          |DiRRAc  |**1.00** $\pm$ 0.00 |
> > |          |Gaussian DiRRAc |1.00 $\pm$ 0.01|
> > | Student Performance| AR   | 0.28 $\pm$ 0.08|
> > |          |MACE    |0.66 $\pm$ 0.12    |
> > |          |ROAR    |1.00 $\pm$ 0.01|
> > |          |DiRRAc  |**1.00** $\pm$ 0.00 |
> > |          |Gaussian DiRRAc |1.00 $\pm$ 0.00|
> >
> > **Minor issues and conclusion section.** Thanks for your comments. we have made the recommended changes to our draft. We have included this paragraph as the Concluding Remarks Section in our paper: “In this work, we proposed the  Distributionally Robust Recourse Action (DiRRAc) framework to address the problem of recourse robustness to model shifts. We introduced a distributionally robust optimization approach for generating robust recourse using a projected gradient descent algorithm. Furthermore,  we also discuss several extensions of our framework. The experiments with synthetic and real-world datasets demonstrated that our framework has the ability to generate recourse that are robust to model shifts under different types of data distribution shifts. We also showed that our framework can be adapted to different model types, linear and non-linear models.”

---

### Official Review · Reviewer_RFkz · 2021-11-04

**Correctness:** 4
**Technical Novelty And Significance:** 2
**Empirical Novelty And Significance:** 2
**Recommendation:** 5
**Confidence:** 3

**Main Review:**

The setting of recourse in the presence of model shifts is well-motivated, especially since models are often updated over time. The idea of formulating the problem as a distributionally robust optimization problem is compelling.

One weakness of this paper is that the technical solution provided is somewhat limited. In particular, the formulation in (4) relies heavily on the structural properties of the mixture distribution and Gelbrich distance to reformulate the optimization problem, and is not surprising given these assumptions. Since this is the main technical result in the paper, it would have been interesting to see a more general setup that considered richer distance metrics.

For the empirical section, a weakness is that DiRRAc (the method proposed in this paper) seems to optimize for ell_2 cost, whereas existing approaches seem to optimize for ell_1 cost. Thus, it is not clear how to compare the costs obtained by DiRRAc with the costs obtained in existing approaches. Moreover, it could be interesting to further investigate the tradeoffs between validity and cost for DiRRAc to illustrate the "cost of robustness" incurred by the approach.

Minor comment:
- Distribut,lly -> "distributionally" p. 4

**Summary Of The Paper:**

The paper provides a framework for recourse (i.e. counterfactual explanations) that is robust to shifts in the model. They formulate the robustified recourse setup as a min-max optimization problem, where the max is over a neighborhood around the distribution over model parameters. The model parameters are drawn from a mixture of K distributions, so that the neighborhood is specified by Gelbrich distance on each component.

They propose a finite-dimensional version of the robustified optimization problem, which can be optimized using projected gradient descent. They evaluate their approach on the German credit dataset, the Small Business Administration dataset, and the Student performance dataset, each of which demonstrates a different type of data distribution shift.

**Summary Of The Review:**

Weak reject due to limited technical contribution and limited empirical comparison

-----
Update after author response: I appreciate the additional experiments for l1 cost included in the revision. Moreover, although I appreciate the author's discussion of the motivation for the Gelbrich distance, I still think that the technical results are somewhat limited. Thus, I keep my assessment the same.

---

> ### Author Response · Authors · 2021-11-22
> **Response to Review**
>
> We thank the reviewer for the comments and suggestions. Our responses are below.
>
> **On the non-triviality of the results.** We would like to clarify that distributionally robust optimization problems are in general difficult because the inner supremum problem is infinite-dimensional. More specifically, for our DiRRAc formulation (2), the supremum problem has a feasible set containing all probability measures that satisfy a certain moment condition. This feasible set is rich because it contains continuous (Gaussian, Laplace, etc.), discrete (binominal, etc.) or even mixed (continuous with jumps) distributions. As a consequence, solving the supremum is hard.
>
> Despite this hardness, our result relies on the Gelbrich distance to compute the exact value of the worst-case probability of mis-classification (Proposition 3.4). To the best of our knowledge, our paper is the first to be able to provide a closed-form expression for this quantity, which depends solely on the model primitives (include the nominal moments and the radius of the set) and does not require solving any other auxiliary problems. Thus, the results in this paper are surprising and non-trivial.
> Moreover, we also add Remark 6.2 to clarify on the necessity of using the Gelbrich distance. Indeed, one may be tempted to try other metrics to prescribe the set. However, it is not possible to arrive at a closed-form reformulation of the DiRRAc problem as we achieved in Theorem 3.2 and in Theorem 4.1.
>
> Choice of Gelbrich distance. In this paper, we choose the Gelbrich distance because it is the only construction of the ambiguity set that leads to closed-form reformulation as in Theorem 3.2 and Theorem 4.1. It is important to note that the Gelbrich distance also possesses many nice performance guarantees both in finite-sample and asymptotic regimes, the details on these guarantees can be found in Theorem 19, 20 and 21 of Kuhn et al. (2019) [1].
>
> **Comparison to other baselines using $l_{1}$ cost.** The following table shows the comparison of our framework with other baselines using $l_{1}$ cost as the optimization metric. We also provide the results with $l_{1}$ cost in our revised draft (Table 1). The costs obtained by DiRRAc and existing approaches are the distances between the original instances and the recourses generated.
>
> | Dataset  |Methods               |$M_{1}$ validity                |$M_{2}$ validity               |$l_{1}$ cost                 |$l_{2}$ cost     |
> |----------|-----------------------------------------|---------------------|---------------------|---------------------|---------------------|
> | German Credit| AR | 0.73 $\pm$ 0.25    | 0.78 $\pm$ 0.00    | 1.26 $\pm$ 0.68    |0.94 $\pm$ 0.41    |
> |          |MACE    |0.87 $\pm$ 0.15|0.97 $\pm$ 0.00    |2.11 $\pm$ 0.86|1.20 $\pm$ 0.47 |
> |          |ROAR    |1.00 $\pm$ 0.00    |1.00 $\pm$ 0.00    |2.60 $\pm$ 0.40    |1.08 $\pm$ 0.16   |
> |          |DiRRAc  |**1.00** $\pm$ 0.00    |**1.00** $\pm$ 0.00|2.09 $\pm$ 0.43    |0.96 $\pm$ 0.18    |
> |          |Gaussian DiRRAc |1.00 $\pm$ 0.00    |0.93 $\pm$ 0.05|**0.73** $\pm$ 0.47    |**0.47** $\pm$ 0.38    |
> | SBA| AR   | 0.26 $\pm$ 0.24 | 0.42 $\pm$ 0.14 | 3.41 $\pm$ 2.10 | 1.56 $\pm$ 0.76 |
> |          |MACE    |1.00 $\pm$ 0.00 |  1.00 $\pm$ 0.00 | 6.85 $\pm$ 0.56 | 2.50 $\pm$ 0.11 |
> |          |ROAR    |1.00 $\pm$ 1.00 |  1.00 $\pm$ 0.00 |   2.25 $\pm$ 0.55 |   0.98 $\pm$ 0.23 |
> |          |DiRRAc  |**1.00** $\pm$ 0.00 |  **1.00** $\pm$ 0.00|    **1.13** $\pm$ 0.43|    **0.82** $\pm$ 0.31 |
> |          |Gaussian DiRRAc |1.00 $\pm$ 0.01|   1.00 $\pm$ 0.00|    1.14 $\pm$ 0.42|    0.83 $\pm$ 0.30|
> | Student Performance| AR   | 0.28 $\pm$ 0.08|  0.35 $\pm$ 0.12|    1.18 $\pm$ 0.99|    0.82 $\pm$ 0.60    |
> |          |MACE    |0.66 $\pm$ 0.12    |0.57 $\pm$ 0.10    |0.81 $\pm$ 0.40|   0.51 $\pm$ 0.24|
> |          |ROAR    |1.00 $\pm$ 0.01|0.98 $\pm$ 0.02|   1.70 $\pm$ 0.27|    0.81 $\pm$ 0.13|
> |          |DiRRAc  |**1.00** $\pm$ 0.00 | **0.99** $\pm$ 0.02| **0.74** $\pm$ 0.18 |**0.63** $\pm$ 0.14 |
> |          |Gaussian DiRRAc |1.00 $\pm$ 0.00|0.98 $\pm$ 0.02|   0.74 $\pm$ 0.18|    0.74 $\pm$ 0.18|
>
> **Cost of robustness of DiRRAc.** In this experiment, we use different values of $\delta$. We define $\delta=\delta_{\min}+\delta_{add}$ with $\delta_{\min}$ is defined in Equation 7. For each value of $\delta_{add}$, we evaluate the validity by using 100 classifiers trained on 100 different data distribution shifts with 3 different shift types. We provide the cost of robustness of DiRRAc (Figure 8) and the comparison with ROAR (Figure 9) in the revised draft.
>
> **Minor issues.** Thanks for your comment, we have revised the paper and corrected the issue in our updated draft.
>
> [1] D. Kuhn, P. Mohajerin Esfahani, V.A. Nguyen, and S. Shafieezadeh-Abadeh. Wasserstein distributionally robust optimization: Theory and applications in machine learning. INFORMS TutORials in Operations Research, pp. 130–169, 2019.

---

### Decision · Program_Chairs · 2022-01-20

**Decision:**

Reject

**Comment:**

The paper provides a framework for recourse (i.e. counterfactual explanations) that is robust to model shifts. The setup for the proposed method is a min-max optimization problem, where the max is over a neighborhood around the distribution over model parameters. The model parameters are drawn from a mixture of K distributions, so that the neighborhood is specified by the Gelbrich distance on each component. The authors propose a finite-dimensional version of the robustified optimization problem, which can be optimized using projected gradient descent. They evaluate their approach on the German credit dataset, the Small Business Administration dataset, and the Student performance dataset, each of which demonstrates a different type of data distribution shift.

Strengths:

- Most existing work on recourse actions do not consider model change, so the problem addressed by the paper is relatively new
- The experiment results demonstrate the superiority of the proposed method over baselines.

Weaknesses:

- The solution provided is somewhat limited as it relies heavily on the structural properties of the mixture distribution and Gelbrich distance to reformulate the optimization problem.

Most of the reviewers voted initially for rejection. The paper is borderline, tending to rejection after the rebuttal. The authors have also considerably updated the paper with new results after the initial reviews. It seems therefore that the paper may benefit from another round of reviewing and, because of this, I recommend rejection and the authors to use the reviewers' comments to improve the paper before resubmitting to another venue for another round of reviewing.